# High throughput 3D gel-based neural organotypic model for cellular assays using fluorescence biosensors

Srikanya Kundu [1], Molly E. Boutin[1], Caroline E. Strong[1], Ty Voss[1] & Marc Ferrer [1✉]

Three-dimensional (3D) organotypic models that capture native-like physiological features of tissues are being pursued as clinically predictive assays for therapeutics development. A range of these models are being developed to mimic brain morphology, physiology, and pathology of neurological diseases. Biofabrication of 3D gel-based cellular systems is emerging as a versatile technology to produce spatially and cell-type tailored, physiologically complex and native-like tissue models. Here we produce 3D fibrin gel-based functional neural co-culture models with human-iPSC differentiated dopaminergic or glutamatergic neurons and astrocytes. We further introduce genetically encoded fluorescence biosensors and optogenetics activation for real time functional measurements of intracellular calcium and levels of dopamine and glutamate neurotransmitters, in a high-throughput compatible plate format. We use pharmacological perturbations to demonstrate that the drug responses of 3D gel-based neural models are like those expected from in-vivo data, and in some cases, in contrast to those observed in the equivalent 2D neural models.

---

[1] 3D Tissue Bioprinting Laboratory, National Center for Advancing Translational Sciences, National Institute of Health, 9800 Medical Center Dr, Rockville, MD 20850, USA. ✉email: marc.ferrer@nih.gov

The development of therapies for neurological diseases has been hampered by the low clinical predictability of both cellular and animal models[1,2]. For decades, conventional two-dimensional (2D) cell culture systems with neuronal cell lines offered simplified and low-cost methods to model brain physiology and drug discovery[3,4]. The advent of human stem cell differentiation technologies has enabled the access to physiological relevant neural cells which are now widely used to study the molecular mechanism of brain and neurological diseases, and as more predictive platforms for drug discovery and development[4,5]. While stem cell-derived cells grown as 2D cultures have been shown to recapitulate many of the cell autonomous features of neurological diseases, these 2D cultures do not capture more complex non-autonomous cell interactions seen in native tissues[6–8]. These complex cell interactions in tissues include local cell-cell cross-talk, interactions between cells and the extra cellular matrix (ECM), local concentration effects due of limited diffusion induced by the ECM, and systemic cell-cell interactions driven by cell secreted factors from other tissues and organs. Due to the lack of these complex interactions, 2D cultures have not been able to reproduce many of the hallmarks of neurological disease as seen in humans. For this reason, in recent years, there has been an increased interest in the development of in vitro 3D organotypic models that capture the native-like physiological cell-cell and cell-ECM interactions seen in tissues and organs. There is now mounting evidence that neural 3D organotypic models mimic the physiology, pathology and pharmacology of human tissues and organs more accurately than the corresponding 2D cellular models. For instance, de Leeuw et al. observed delayed expression of tau protein and differentiation and maturation of a network of iPSC-derived cortical neurons in 2D culture compared to 3D, thus limiting the development of disease features of Alzheimer's disease (AD) and other 'tauopathies' in vitro[9]. In another study, Lee et al. showed that 3D neurons showed less reduction of Aβ compared to 2D neurons at the same concentrations of BACE1 or γ-secretase inhibitors, illustrating how in some instances, the pharmacological effects are only seen in 3D organotypic models with the appropriate physiological complexity[10]. It has also been shown that addition or enhancement of ECM deposition in neural 2D co-cultures accelerated the formation of neural networks, and in one example, the drug responses were more similar to those obtained in vivo[11,12], thus reproducing evidence from in vivo experiments that ECM plays an important role in the development and function of the neural circuitry in the brain. Similar data has now also been shown for many non-neuronal 3D tissue models[13–15].

3D organotypic neural models range from spheroids/organoids/assembloids[16–21], scaffold-based engineered systems[22], and tissue- and organ-on-chip platforms[23], each providing different levels of cellular complexity, physiological, morphological, and mechanical features, as well as ease of scale-up and use as robust and reproducible platforms for drug discovery and development[24]. The development of brain organoids which are generated from iPSC using guided differentiation[16,17] have produced 3D organotypic neural models with relevant cellular complexity and some native-like tissue morphologies and have provided insights into brain development in normal and disease states. However, organoids are heterogenous in nature, with low level of reproducibility in size, cell maturation and composition, and limited physiological features, and do not have the reproducibility needed to make them robust drug testing platforms. Tissue engineering techniques are enabling the assembly of tissue-like 3D organotypic models with tailored cell type and ECM composition, controlled spatial arrangements, intricate physiological features such as vasculature, and with the reproducibility and robustness necessary to allow medium throughput drug screening. Biofabrication techniques are being used to create 3D scaffold-based neural co-cultures models using human stem cells derived neurons and with defined spatial arrangements that created functional neural connectivity, as measured by calcium waves or electrophysiology with microelectrode arrays[25,26]. These biofabricated neural models show promising improvement in terms of versatility in physiological and morphological features, and sophisticated directed spatial orientation of neuronal arrangements with long neuronal connectivity[27]. These bioengineering approaches have also been used to improve the formation of organoids models by including relevant hydrogels, patterning, and flow[28]. However, in general, these biofabricated 3D organotypic models, while they include many relevant and sophisticated physiological and morphological features, they are quite complicated to assemble, normally as a single use device and not in a multiwell plate format, and require the prototyping and fabrication of devices in specialized laboratories, thus limiting their large scale adoption by the larger scientific community, especially for drug screening applications that require medium to high throughput. There is therefore the need for easy to use, versatile and modular bioengineering methods to rapidly create neuronal culture models that have native-like neuronal densities, tailored cell type composition, controlled spatial arrangements to generate local and non-proximal neuronal circuits mimicking those in the brain, and amenable to physiologically relevant functional assays in a multiwell plate format so that they are readily used for drug testing in a high throughput screening format.

Towards the goal of generating robust and functional biofabricated neuronal cellular models with tailored cell type composition, controlled spatial deposition, and that are amenable for drug testing, we have used human iPSC differentiated neurons and astrocytes with a soft gel-based extracellular matrix that enables deposition of viable cells with defined cellular composition. The neurons are cultured at native-like neuronal densities and allowed to form functional neurites and autonomous synaptic connections. In addition, functional assays were developed by introducing genetically encoded fluorescence biosensors for calcium activity and levels of neurotransmitter to enable real time measurements of basal signals, optogenetically evoked states, and external pharmacological perturbations. We have developed a 3D gel matrix consisting of a fibrin-based polymeric mesh with gelatin and laminin with the optimal stiffness and adherent properties to enable the co-culture of viable human iPSC derived dopaminergic or glutamatergic neurons with astrocytes, and formation of a network of long and extended neurites with active synapses. For the development of functional assays, we performed AAV-mediated double transfection of genetically encoded ChrimsonR-opsin[29] with combination with different biosensors, GCaMP6f[30], dLight1.2[31] and iGluSnFr[32] for measurements of intracellular calcium flux, and released neurotransmitters, dopamine, and glutamate, respectively. This fibrin gel-based 3D neuronal co-culture models proved to be functional by measurements with the different biosensors and responded to optogenetics stimulation[33]. A set of compounds that target relevant neuronal receptors was used to show that pharmacological perturbations on the 3D neural cellular models produced functional responses as to be expected from in vivo data, in some cases contrasting those obtained in a 2D co-culture counterpart. These 3D neural organotypic models were produced in a 384-well platform, in a reproducible manner to be amenable for scale up and therapeutics testing. Furthermore, the composition of gel-matrix used here is compatible with extrusion-based bioprinting and should enable the biofabrication of spatially defined functional neuronal circuits. Finally, the approach is very modular and allows for inclusion of additional physiological features, including the use of alternative more relevant hydrogels like human brain tissue-derived brain extracellular matrices[28] and the addition of other relevant cells like microglia and vasculature[34].

## Results

**Production and characterization of fibrin gel-based 3D neural organotypic model on multi-well microplates.** In the present study, we sought out to produce engineered, gel-based 3D neural co-culture models from iPSC derived cells with spontaneously formed functional neuronal networks, and with the ability to modulate network activity to mimic the human brain neuronal activity. We constructed two neuronal co-culture models in 2D and 3D format using iPSC-derived dopaminergic neurons (iDopas) with iPSC-derived astrocytes (iAstros), and iPSC-derived Glutamatergic neurons (iGlutas) with iAstros, on 384-well microtiter plates (Fig. 1a). The schematic of the overall experimental approach for the construction of the functional 2D and 3D neural co-culture models, expressing relevant fluorescent biosensors, on multi-well format is shown in Fig. 1b. The mature 3D gel-based neural co-cultures were approximately 100–200 μm in thickness, as determined by z-stacking brightfield confocal imaging with several neuronal markers (Fig. 1c, middle panel), while the 2D co-cultures were 3–5 μm thick. After 2 weeks in culture, the 3D neural tissues showed axonal projections to form a mesh-like structure, which was notably lacking on the 2D model, as measured by brightfield macroscopy (Fig. 1c, 3D rendered image in middle panel and Supplementary Movie 1). 3D rendering and reconstruction of z-stacks brightfield confocal images (pseudo color orange) revealed that the fibrin-based gel matrix provided the suitable stiffness to allow the axonal projection to grow horizontally and to hold the mesh vertically as well (Fig. 1c, middle panel, Supplementary movie 1). Cell viability assessment using CalceinAM and Propidium iodide staining confirmed that there were no major differences in the live and dead cells ratio between the 3D and 2D co-culture models. The 3D model had 76% live and 24% dead cells, statistically significant confirmed by $t$-test ($p < 0.0001$) whereas in 2D system average live and dead cells are 79 and 21% ($t$-test significance $p < 0.0001$) (Fig. 1c bottom panel and bar graph). We also tested the cell viability of our 3D model using an ATP-based 3D cell viability assay (Supplementary Fig. 1) and uncovered that the addition of 1 μg/ml of laminin helped promoting the axonal growth as literature suggested[35,36] into our 2.5 mg/ml of fibrinogen gel, also improved the viability of our neuronal coculture system as a whole by almost 2-fold ($t$-test $p < 0.01$ in Supplementary Fig. 1b).

To evaluate the neuronal integrity and network formation in the human iPSC-derived neural co-culture models, we performed immunostaining-based fluorescence microscopy analysis on relevant biomarkers of cell identity and neuronal synapse. The confocal images of 2D co-cultures and the 3D rendering of z-stack images from 3D models are shown in Fig. 2. Identification of dopaminergic and glutamatergic neurons was assessed by the antibody labelling (green color) with TH, a protein selective for developing dopaminergic neurons[37] on the iDopas/iAstros co-cultures, and with VGLUT1 protein for glutamatergic neurons[38] on the iGlutas/iAstros co-cultures. Both 2D and 3D iDopas/iAstros (Fig. 2a) and iGlutas/iAstros (Fig. 2b) co-culture models showed that almost 90% Hoechst nuclear stained cells were positive for the corresponding neuronal marker. Expression of PSD95 markers (yellow), often overlapping on the neurites/cell bodies and alongside at the neuronal junction's was used to assess formation of synaptic connections (Fig. 2a and b). Staining of MAP2 was used to assess the formation of a neurite network and GFAP showed the population of astrocytes in the iDopas/iAstros (Fig. 2c) and iGlutas/iAstros (Fig. 2d) models in 2D and 3D format.

Furthermore, we designed multiple image-based physiologically relevant functional assays based on the expression of genetically encoded fluorescence biosensors coupled with high content imaging. The experimental approach used for these assays is shown in Fig. 3a. z-stack confocal fluorescence imaging with 3D rendering and image reconstruction showed that expression pattern of each biosensor was different and as expected: GCamp6f expressed on the neural body and throughout the cell membrane (white arrows, Fig. 3b′); dLight1.2 appeared as a cluster of puncta at the synaptic ends (yellow arrows in Fig. 3b′′); and iGluSnFr traced along the axonal length (yellow arrows in Fig. 3b′′′) (see also Supplementary Movies 1 and 2).

**Dynamic intracellular calcium measurements with genetically encoded GCaMP6f biosensor.** To assess whether viral-genome transfection and expression of the biosensors would alter the physiological calcium activity in the co-cultures, we first compared calcium signal with Calbryte 520AM and GCaMP6f biosensor in 2D iDopas/iAstros neural co-culture system, using a well-based FLIPR fluorescence reader. Examples of acquired calcium fluorescence traces are shown in Supplementary Fig. 3a and c. Repetitive calcium peaks were detected for both the Calbryte 520AM and GCaMP6f assays over the period of 600 sec, suggesting that this 2D neural co-culture produced active and synchronized neuronal networks, and that expression of GCaMP6f via AAV transfection did not alter the neural calcium activity trend significantly (Supplementary Fig. 3a and c). Further functional validation with pharmacological interventions showed the equivalent effects of apomorphine, a D2-receptor agonist, in both assays. Apomorphine decreased the calcium peak frequency in both calbryte520AM dye and GCaMP6f transfected assay systems ($p > 0.001$ and $p > 0.0001$, respectively) with no significant differences in amplitude (Supplementary Fig. 3b and d). Apomorphine showed a decrease in average synchronized peak frequency from 1.87 to 0.7 for the dye-based assay, and from 0.76 to 0.42 for the biosensor assay.

While it was possible to measure calcium activity with the GCaMP6f and Calbryte on the 2D neuronal co-culture models using a well-based FLIPR reader, we were not able to detect signal from the 3D co-culture models using the same approach. In addition, the FLIPR instrument did not allow for optogenetics stimulation. These motivated to design fluorescence microscopy-based calcium detection protocols (Fig. 3c) which could work on both the 2D and 3D gel-based systems, and enabled stimulation of the genetically encoded ChrimsonR-opsin, in combination with calcium activity measurements. We validated this protocol using two different neural co-culture systems: one included iDopas/iAstros (Fig. 4a–f) and another iGlutas/iAstros (Fig. 4g–l). Figure 4a–d show representative GCaMP6f fluorescence images of unstimulated (basal) and light-stimulated (evoked), with and without treatment with apomorphine, the D2 receptor agonist, taken from 2D and gel-based 3D neural iDopas/iAstros co-cultures, using epifluorescence and confocal (one z-plane) modes, respectively. The expression of ChrimsonR-opsin was confirmed by td-Tomato (red) fluorescence (Supplementary Fig. 4a, b). Representative traces of calcium signal via GCaMP6f intensity, normalized by their mean fluorescence intensity ($dF/F_0$) from three randomly picked neurons were shown in Fig. 4b and e, for 2D and 3D co-culture models, respectively. Cumulative data on peak frequency and peak amplitude from each experimental group are summarized as bar plots in Fig. 4c for 2D and in Fig. 4f for 3D model. The data indicate that the optical stimulation significantly enhanced cellular calcium peak frequency by approximately 2-fold for both 2D ($p < 0.0001$) and 3D ($p < 0.0001$) models. Though the average evoked peak amplitude did not significantly change in 2D, it increased by ~50% in the 3D system ($p = 0.0042$). In the 3D system, apomorphine treatment decreased the basal peak frequency by 40% ($p = 0.01$) and amplitude by 60% ($p = 0.0049$), and the evoked peak frequency by 40% ($p < 0.001$), but there was no

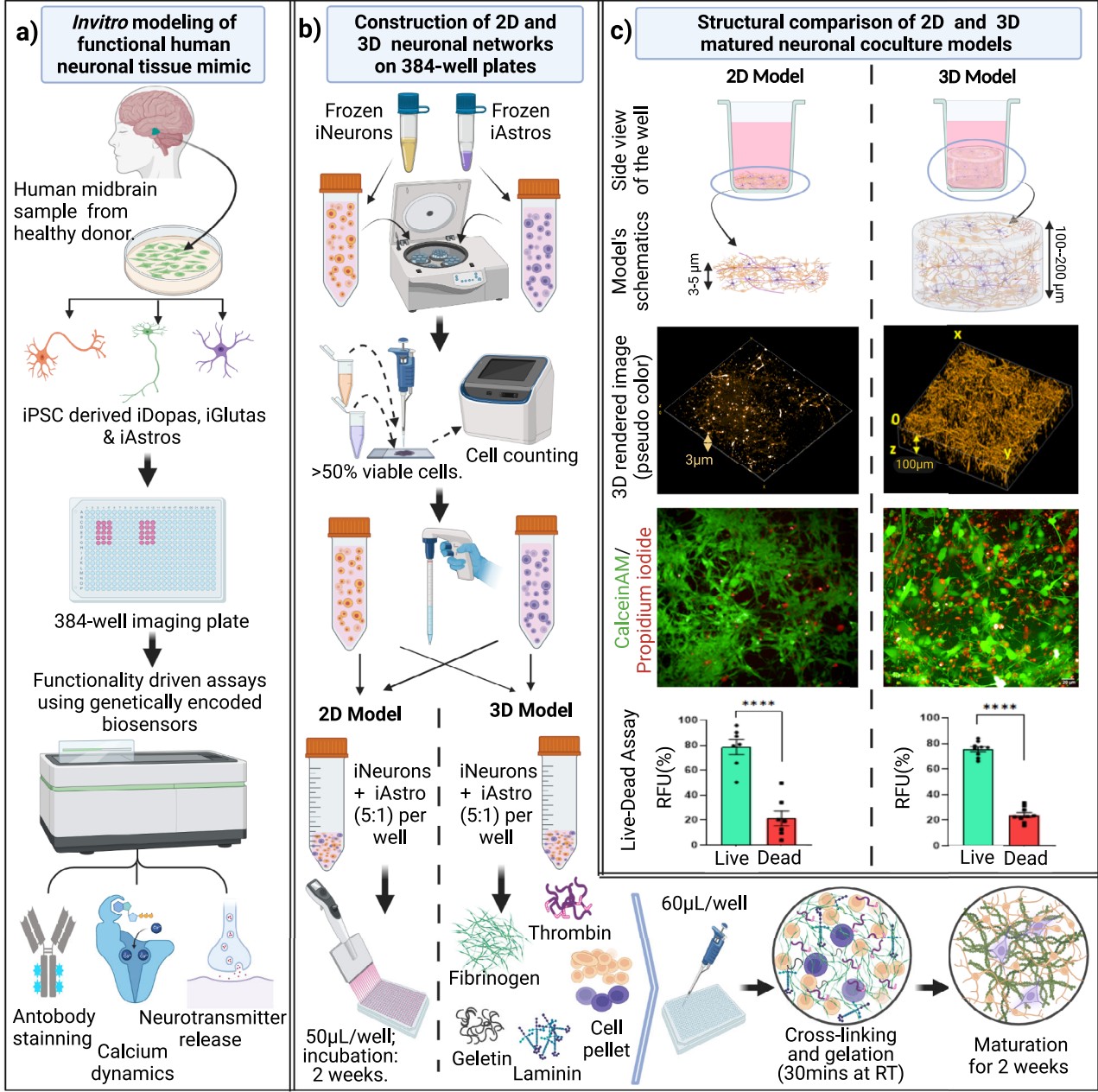

**Fig. 1 Schematic for the biofabrication of functional 3D gel-based neural co-culture cellular models for high content screening in a multi-well plate platform. a** The neural coculture models were developed using iPSC derived post-mitotic midbrain dopaminergic or glutamatergic neurons with iPSC-derived astrocytes from healthy human donor in 384 well flat bottom imaging plates. We designed three functionality driven assays with spatiotemporal resolution using genetically encoded biosensors: (i) Measurement of single cellular level 'in-network' calcium dynamics over time with CAG-GCaMP6f sensor. (ii) Assessment of released dopamine neurotransmitter at the synaptic cleft with hSyn-dLight1.2 sensor. (iii) Detection of glutamate neurotransmitter within network with hSyn-GluSnFr sensor. **b** Flow chart for the assembly 2D and 3D gel-based neural models. Frozen human iPSC derived neurons and astrocytes were thawed, counted, and mixed at the desired ratios. For 2D co-cultures, plates were coated with PLO-laminin and 50 μl of the cell suspension in media was pipetted into the wells. For 3D gel-based co-cultures, 60 μl of cell suspension in a mixture of fibrinogen (2.5 mg/ml), gelatin (60 mg/ml), laminin (1 mg/ml) and thrombin (1:1000U) in media were quickly pipetted in the wells before gelation. **c** After gelation and maturation, the 3D fibrin gel structure was 100–200 μm thick compared to 3–5 μm thin 2D co-cultures. After 14 days, a z-stack of confocal brightfield images (pseudo-orange color) from 3D system indicates horizontal as well as vertical progression of neurites through the gel matrix, whereas the images from 2D system (pseudo-orange color) does not show any downward spreading of neurites. The calceinAM/propidium iodide live-dead imaging assay confirmed the presence of that 80% of the cells were live cells in both models. Error bar s.e.m., three biological replicates, each $n = 3$. Scale bar 20 μm. Schematics made in BioRender.

statistical change in the amplitude (Fig. 4f). Apomorphine did not induce any statistical changes in neither basal peak frequency nor in amplitude but showed a 1.7-fold increase in evoked peak frequency ($p < 0.001$) and no significant changes in peak amplitude in 2D (Fig. 4c).

The same measurements were done in a 3D fibrin-gel neuronal model of iGlutas/iAstros, as described in Fig. 3. For the iGlutas/iAstros co-culture system, D-serine, an NMDA receptor agonist, was used for studying pharmacological modulation (Fig. 4g–l). The representative single plane images, corresponding single

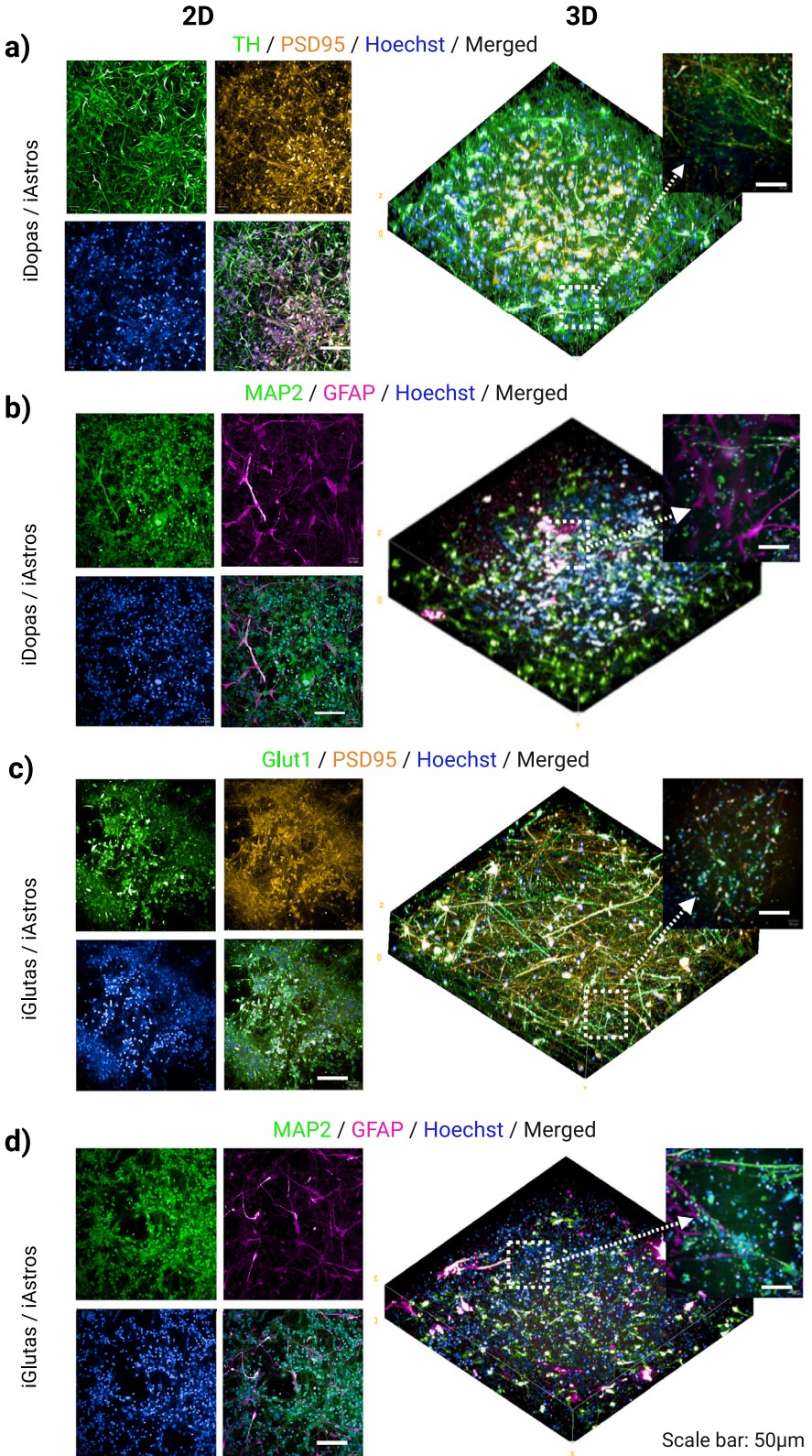

neuronal calcium traces and cumulative data on peak frequency and peak amplitude from basal, basal after D-serine treatment, evoked and evoked after D-serine treatment groups are shown in Fig. 4g–i and 4j–l for the 2D and 3D models, respectively. The corresponding td-Tomate fluorescence images of ChrimsonR expression are shown in Supplementary Fig. 4c and d for 2D and

3D system, respectively. D-serine treatment increased the basal calcium peak frequency 2-fold for the 2D ($p = 0.0124$) and 3-fold for 3D models ($p < 0.0001$); and elevated calcium peak amplitude by 3-fold in the 2D culture ($p = 0.0001$), but there was no significant increase in peak amplitude in the 3D culture. As expected, optogenetic stimulation elevated peak frequency by

**Fig. 2 Detection of neural markers by immunostaining and fluorescence microscopy in 2D and 3D iPSC derived neuron/astrocyte co-cultures models.**
Post PFA fixation, neural 2D and 3D co-culture models stained with antibody-based neural receptor selective identity markers and makers of neuronal network. The representative confocal microscopic images (10x) are shown as quadro-tiles format for 2D and merged three-dimensional image of post 3D rendering and reconstruction for 3D model. **a** Dopaminergic neurons and astrocyte cocultures were double immune-stained with Tyrosine Hydroxylase (green) and post synaptic marker PSD-95 post-synaptic (yellow) antibodies, and Hoechst 33342 nuclear stain (blue). **b** The glutamatergic neurons and astrocytes cocultures were double-stained with Glut1 transporter (green), PSD-95 post-synaptic (yellow) antibodies and with Hoechst 33342 nuclear (blue). **c, d** For determination of neuronal branching, co-cultures were stained for MAP2 (green) and with the astrocyte marker GFAP (purple). The 3D reconstruction view confirmed the homogeneous distribution of neuronal cells and their neurites along with astrocytes. The insets (40x confocal) show the closeup of markers expressions on respective targets. Scale bar 50 μm; inset's scale bar 20 μm.

about 2-fold in both 2D ($p = 0.0002$) and 3D ($p < 0.0001$) but did not produce a statistically significant increase in peak amplitude for neither 2D nor 3D cultures. D-serine induced a reduction of ~50% in the evoked peak frequency in the 2D system ($p = 0.0131$) but had no effect on this measure for the 3D model. D-serine had no effect on evoked peak amplitude for neither the 2D nor 3D co-culture models.

To further establish the pharmacological perturbation on the 2D and gel-based 3D iDopas/iAstros and iGlutas/iAstros coculture systems on the intracellular calcium signal, we tested the effect of additional compounds that modulate different relevant neural targets: metoclopramide, a D2-receptor antagonist, ketamine, an NMDA-receptor antagonist, and morphine, a μ-opioid receptor agonist. During process of validating the neuronal cultures, we detected expression of μ-opioid receptor (MOR) protein marker (Supplementary Fig. 5) on the cell bodies (at the proximity of Hoechst staining) on both of our 2D and 3D gel-based models, which prompted us to investigate the pharmacological effects of an μ-opioid receptor agonist on our functional neuronal assays.

Figures 5a and 5b show calcium traces of randomly picked neurons from the iDopas/iAstros 2D and 3D co-cultures systems, respectively, under each experimental conditions: basal, basal after respective drug treatment, evoked and evoked after respective drug treatment. Average peak frequency and peak amplitude are represented as bar plots in Fig. 5c–f, for 2D and 3D co-cultures systems, respectively. Our data indicate that pharmacological targeting of different neuronal receptors produced different modulation of calcium activity in the 2D vs 3D co-cultures. For instance, metoclopramide, the D2-receptor antagonist, reduces basal peak frequency by half ($p = 0.0062$) in the 2D cellular models but increases it from 0.062 to 0.1 pps ($p < 0.0001$) in 3D gel-based system. On the other hand, metoclopramide decreased evoked peaked frequency in 2D from 0.4 to 0.29 pps ($p = 0.0005$), but it did not have a significant change evoked peaked frequency in the 3D model. D-serine induced a decrease of ~0.1 pps ($p < 0.0001$) on basal peak frequency but a modest increase from 0.379 to 0.458 pps ($p < 0.001$) on evoked calcium activity in 2D co-cultures; in contrast, on 3D co-cultures, we observed an increased peak frequency in both basal (from 0.132 to 0.166 pps, $p < 0.001$) and evoked (from 0.193 to 0.24 pps, $p < 0.0001$) calcium signal after D-serine treatment. Ketamine treatment suppressed basal peak frequency half in 2D ($p = 0.0009$), whereas it increased basal peak frequency from 0.136 to 0.209 pps ($p < 0.0001$) in the 3D model. No significant effects were observed by ketamine treatment in the evoked peak frequency in neither the 2D vs 3D cellular models. Finally, treatment with morphine produced significant suppression in both basal and evoked peak frequency (from 0.288 to 0.209 pps, $p = 0.0039$ and over 4.5 folds, $p < 0.0001$ respectively) in 2D co-culture. In contrast, in 3D gel matrix, morphine induced a significant increase in basal (from 0.145 to 0.259 pps, $p < 0.0001$) as well as evoked (from 0.213 to 0.261 pps, $p = 0.0006$) peak frequency.

The effects on calcium peak amplitude were also analyzed: the treatments with the different drugs did not have any significant effects on the basal nor evoked peak amplitude on 2D co-cultures (Fig. 5e). On the other hand, for the 3D gel model, there was significant modulations with calcium peak amplitude (Fig. 5f). We observed decrease after metoclopramide treatment in basal peak amplitude (from 2.75 to 1.286 dF/$F_0$, $p = 0.0079$) but an increase (from 0.878 to 3.28 dF/$F_0$, $p < 0.0001$) after optical stimulation. D-serine increases basal peak amplitude significantly (from 0.462 to 2.31 dF/$F_0$, $p < 0.0001$) without modulating evoked peak amplitude. Finally, morphine suppressed (from 1.783 to 1.13 dF/$F_0$, $p = 0.0368$) the basal peak amplitude but not the evoked one.

Representative traces of iGlutas/iAstros co-cultures from each experimental group from 2D and 3D models are shown in Fig. 6a and b, respectively. Different changes in calcium peak properties for each drug treatment between 2D and 3D co-culture models are shown in Fig. 6c–f. In the 2D cellular model, ketamine reduced basal and evoked calcium peak frequency by 40% ($p = 0.0477$) and 60% ($p < 0.0001$), respectively. In the 3D co-culture system, ketamine did not produce a statistically significant decrease in basal peak frequency, but it did induce a significant 5-fold decrease ($p < 0.0001$) in evoked peak frequency. Our data also indicated that the D2 receptor modulators apomorphine and metoclopramide did not have significant effects on iGlutas/iAstros coculture basal peak frequency, neither in 2D nor in 3D models. However, they modulated the evoked calcium peak frequency: apomorphine had no significant effect on evoked peak frequency in 2D but decreased evoked peak frequency by 5-fold ($p > 0.0001$) in the 3D co-culture model. Metoclopramide significantly decreased evoked peak frequency by 60% ($p > 0.0001$) in 2D but it increased by 60% ($p > 0.0001$) in 3D cultures. Morphine also exhibited contrasting results: in the 2D coculture systems, it did not modulate the basal peak frequency, but it significantly suppressed the evoked peak frequency by 40% ($p = 0.0002$). In 3D co-cultures, morphine increased basal peak frequency by 2-fold ($p = 0.007$) but had not significant effect on the evoked peak frequency. When assessing effects on calcium signal peak amplitude, there were no significant changes in the 2D basal peak amplitude for any of the compounds tested, and there were some effects on the evoked peak amplitude with significant decreases from 0.687 to 0.197 dF/$F_0$ ($p < 0.0001$) by metoclopramide and significant increases of ~2-fold for ketamine ($p = 0.0047$) and from 0.183 to 0.4 dF/$F_0$ for morphine ($p = 0.0398$). For the 3D models, there were significant effects on basal peak amplitude with apomorphine (3-fold decrease, $p < 0.0001$), metoclopramide (from 3.5 to 5.8 dF/$F_0$ increase, $p = 0.0017$), ketamine (2-fold increase, $p < 0.0001$). However, in the 3D cultures, none of the treatments had any significant effects on the evoked peak amplitude.

Single cell calcium dynamics traces from randomly picked wells from the apomorphine treated group of iDopas/iAstros co-cultures after a wash-out period of 24 h are shown in Supplementary Fig. 6. For both 2D and 3D iDopas/iAstros

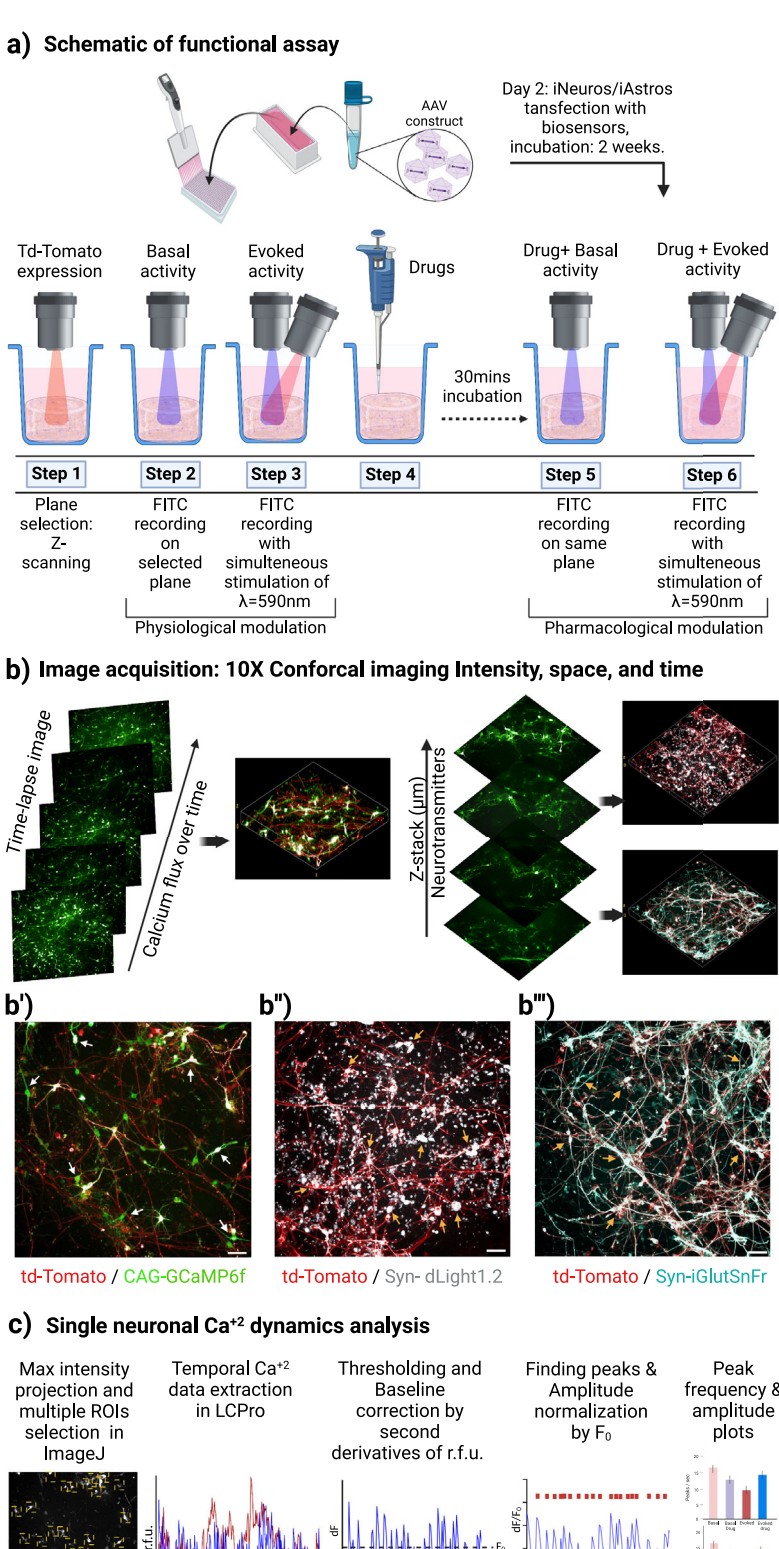

**a)** Schematic of functional assay

**b)** Image acquisition: 10X Conforcal imaging Intensity, space, and time

**b')** td-Tomato / CAG-GCaMP6f

**b'')** td-Tomato / Syn- dLight1.2

**b''')** td-Tomato / Syn-iGlutSnFr

**c)** Single neuronal Ca$^{+2}$ dynamics analysis

co-cultures, calcium activity was restored to pre-treatment levels after 24 h of apomorphine washout (Supplementary Fig. 6a and c respectively). Furthermore, similar evoked calcium activity in terms of frequency and amplitude (Supplementary Fig. 6b and d) was observed for both the 2D and 3D iDopas/iAstros co-cultures, after 24 h of apomorphine washout.

**Image-based functional assays for the detection of released neurotransmitters**. Next, we developed image-based functional assays to quantitate the release of neurotransmitters from both neural co-culture networks. Expression of ChrimsonR opsin was confirmed by td-Tomato fluorescent (Fig. 7a, first column). Representative images of dLight1.2 intensities from 2D and 3D

**Fig. 3 Schematic flow diagram for the development of image-based functional assays using genetically encoded biosensors and analysis of single cellular 'in-network' calcium dynamic from 2D and 3D neural co-culture models.** (i) Genetically encoded biosensor AAV9-CAG-GCaMP6f allowed over time live calcium flux measurement, (ii) AAV5-hSyn-dLight1.2 biosensor captured real time quantitation of synaptic released of dopamine neurotransmitter with spatial resolution (multiple fields of view through the z-stack), (iii) detection of glutamate neurotransmitter was done with a AAV9-hSyn-iGluSnFr biosensor with spatial resolution (multiple field view through the z-stack). Controlled physiological modulation with AAV-hSyn-ChrimsonR-td-Tomato transfection and optogenetic stimulation ($\lambda = 590$ nm) added functional complexity to the 3D system. **a** The progressive development of assays was demonstrated here. The ChrimsonR opsin in combination with one of the biosensors were transfected at culture day 2 and allowed to incubate for 2 weeks. On the analysis day, first the imaging plane/s were determined by the td-Tomato expression by scanning the entire z-stack. Then the spontaneous activity of the neuronal network for all three biosensors as green fluorescence were captured using a FITC filter ($\lambda_{Ex} = 495$ nm; $\lambda_{Em} = 519$ nm). These steps were repeated with simultaneous stimulation with red ($\lambda = 590$ nm) light for evoked network activity detection. Next, the co-cultures were treated with receptor specific pharmacological agents for 30 min and both the basal and evoked acquisition process were repeated sequentially for the pharmacological perturbations. **b** The image acquisitions was performed under 10X air epifluorescence for 2D and 10X water confocal microscope for 3D system. Both the basal and evoked activity from the neuronal network was recorded either over time (180 s with 0.6 s acquisition interval) from a selected plane for GCaMP6f biosensor, or green fluorescence intensity were acquired from multiple fields of views over the entire z-stack for dLight1.2 or iGluSNFr biosensors. 3D rendering images of all three biosensor's signals along with ChrimsonR expression via td-Tomato fluorescence, from 3D fibrin gel neuronal model are presented alongside. Scale bar 50 μm. **b'**, **b''** and **b'''** Corresponding enlarged z-stack images of each biosensor with td-Tomato expression are presented here. The arrows indicate the fluorescence of the biosensors on the neural morphology. Scale bar 50 μm. **c** The single neuronal calcium data analysis from time lapsed GCaMP6f images was performed as followed. A series of 300 time-lapsed images were captured over a period of 180 s with 0.6 s interval by Harmony v5.1 acquisition software. The images were stacked to provide maximum intensity projection and transferred to 'ImageJ' image processing software. Then, ROIs were automatically selected centering on the neuronal soma by the 'LC-Pro' plug-in in 'ImageJ' by providing the ROI diameter (pixel) 30, frame rate (fps) 1.6 and for intensity cutoff threshold with $p$-value 0.05. Finally, the raw intensity values of the calcium signal were extracted from selected ROIs. For the analysis, each ROIs data was then transferred to 'Origin-Pro 9.0'. The peaks were detected over time by 'positive maximum intensity peak finding method' using 'batch peak processing algorithm', with second derivatives of individual ROI's raw fluorescence intensities (r.f.u.) as a change in fluorescence intensity from initial (dF), because of sparsity of event's appearance in our 3D system. After thresholding the base line, the estimation of initial fluorescence level ($F_0$) for each ROIs was calculated by averaging the r.f.u. and then the peak amplitude was normalized to $dF/F_0$ for respective ROIs. We did not used initial image frame's mean fluorescent intensity as our $F_0$, because in both of 2D and 3D model on every image frame we had GCaMP6f activated fluorescence intensity from cells to some extent. The oscillations of calcium waves ($dF/F_0$) of 3–4 example traces from each experimental group were presented and single-neuron dynamics were quantified by peak frequency (peaks per sec) and peak amplitude ($dF/F_0$) measurement. Schematics made in BioRender.

model of iDopas/iAstros co-culture with the different treatments are shown in Fig. 7a (green). Quantitation of green-fluorescent intensity from the dLight1.2 dopamine biosensor at basal, basal with drug treatment, evoked and evoked drug treatment are shown in Fig. 7b and Fig. 7c, for 2D and 3D co-culture systems, respectively. Both cellular models showed significant increase of dopamine release upon optogenetic stimulation, 3-fold, ($p > 0.001$) for 2D and 2-fold ($p > 0.0001$) for 3D co-cultures, respectively. Apomorphine significantly decreased the basal release of dopamine levels by 2-fold ($p = 0.021$) in 3D gel-matrix but had no significant effects on 2D co-cultures. Apomorphine also reduced the evoked release of dopamine in 3D by 3-fold ($p < 0.0001$) more effectively than in 2D (by 30%, $p = 0.0438$). For 3D co-cultures model, our results showed an increase in the basal release of dopamine after metoclopramide (>2-fold, $p = 0.0001$), D-serine (2-fold, $p = 0.0394$) and morphine (2-fold, $p = 0.0011$) treatment, but no statistically significant effects with ketamine treatment. In contrast, in our 2D co-culture, there was no statistically significant changes on basal dopamine release after metoclopramide, D-serine and ketamine treatment but there was a statistically significant increase in after morphine (from 11.46 to 26.56 r.f.u., $p < 0.0001$) treatment. As expected, we observed significant increase in dopamine neurotransmission after optogenetic stimulation in both of our models on each drug modulation experiments. For instance, in our first group we observed increased in fluorescence intensity from 12.72 to 37.41 ($p < 0.0001$) and from 13.8 to 29.06 r.f.u. ($p < 0.0001$) for 2D and 3D co-cultures, respectively. In these evoked experiments, metoclopramide further enhanced the dopamine release in both of our 2D (from 40.006 to 60.44 r.f.u., $p > 0.0001$) and 3D (from 27.29 to 57.29 r.f.u., $p > 0.0001$) model. For all other drugs effect, the evoked release of dopamine was reduced in the 3D gel-based model: D-serine (from 37.62 to 27.05 r.f.u., $p = 0.0485$), ketamine (from 40.62 to 16.13 r.f.u., $p < 0.0001$) and morphine (from 39.35

to 29.28 r.f.u., $p = 0.001$). In contrast, in 2D cultures, we recorded no statistically significant changes in evoked dopamine release after D-serine and morphine treatment but significant decrease after treatment with ketamine (from 33.85 to 15.9 r.f.u., $p < 0.0001$).

Next, we sought to detect glutamate release from the neural co-culture systems. Since glutamate can be released from glutamatergic neurons and activated astrocytes, and astrocytes take part in glutamate trail from the system via glutamate transporters as well[39], we used the glutamatergic neuron and astrocyte co-culture for this assay. We confirmed the expression of ChrimsonR opsin by acquiring td-Tomato fluorescence image from both co-culture models (Fig. 8a, first column). Unlike the green puncta observed for d-Light1.2, the dopamine sensor, the iGluSnFr sensors appeared longitudinally along the neurites length and also concentrated at the neuron-neuron or neuron-astrocyte junctions (Fig. 8a). Quantitation of green-fluorescent intensity from the iGluSnFr glutamate biosensor at basal, basal with drug treatment, evoked and evoked drug treatment are shown in Fig. 8b and c, for 2D and 3D co-culture systems, respectively. We observed a significant increase in the basal glutamate green fluorescence over 2-fold, from 6.15 to 15.01 r.f.u. ($p = 0.0085$) in 3D but no statistically significant increase in 2D co-culture models after D-serine treatment. Upon optogenetics stimulation, no significant effects of D-serine were detected on the evoked glutamate levels on 2D co-cultures, and significant increases were observed in evoked glutamate signal (from 22.94 to 33.13 r.f.u. $p = 0.0062$) in the 3D co-cultures. The acquired data showed that there were no statistically significant differences in basal or evoked glutamate level after the treatment with apomorphine and metoclopramide in either 2D or 3D co-cultures. ketamine produced a statistically significant reduction in glutamate release in both basal (3-fold, $p = 0.0021$) and evoked (>2-fold

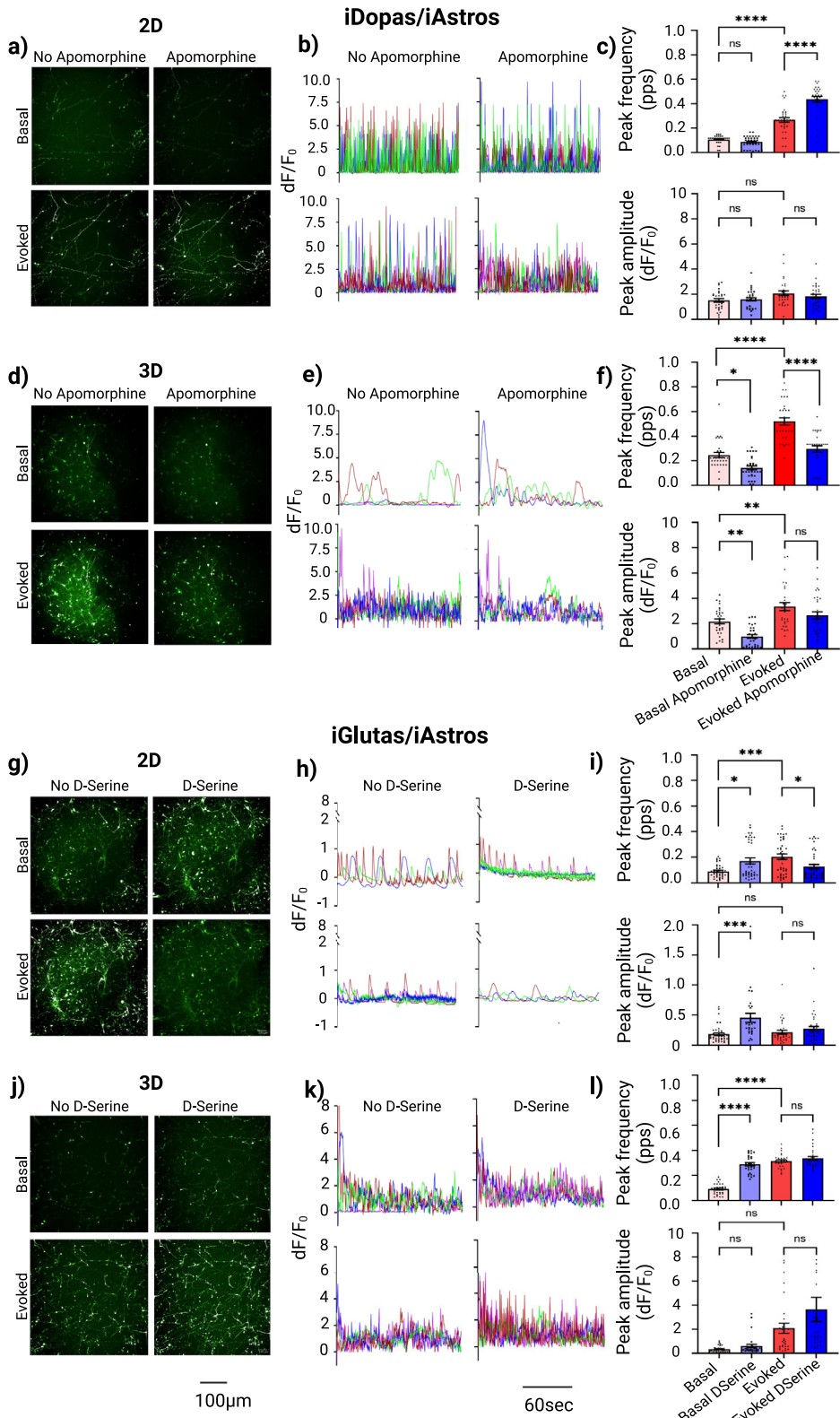

$p < 0.0001$) for 2D, but only 1.5-fold ($p = 0.01$) reduction in evoked released in 3D. We did not observe any changes in glutamate level before and after morphine treatment at basal or evoked conditions for the 2D model but observed an increase (from 9.01 to 15.16 r.f.u., $p = 0.051$) for basal release of glutamate in the 3D system.

## Discussion

The use of human iPSC is enabling the development of in vitro cellular models of the human brain which until very recently were not possible to produce. It is now possible to study brain developmental biology, cellular mechanisms underlaying neurological and neurodegenerative diseases in vitro and using these cellular

**Fig. 4 Measurement of single cell calcium dynamics using genetically encoded GCaMP6f biosensors in iDopas/iAstros and iGlutas/iAstros 2D and gel-based 3D co-cultures.** The modulation of network calcium dynamics with the D2-receptor agonist apomorphine (10 μM) for iDopas/iAstros and the NMDA-receptor agonist, D-serine (10 μM) for iGlutas/iAstros was quantitated before and after optical stimulation measuring peak properties including peak frequency per sec and peak amplitude. **a** and **d** are representative images from pre and post apomorphine treatment and **g** and **j** are representative images from pre and post D-serine treatment of basal and evoked calcium activity. Scale bar 50 μm. **b** and **e** and **h**, **k** Example calcium traces over time were plotted for 2D and 3D co-cultures, for iDopas/iAstros and iGlutas/iAstros respectively. **c**, **f**, **i** and **l** The peak frequencies (peak per sec) and mean peak amplitudes from each ROI, minimal of 3 ROIs from each well and three biological replicates, each with 3 wells per group were plotted as bar graph. Error bar are s.e.m., Statistics: Two-way ANOVA with post hoc turkey test between groups.

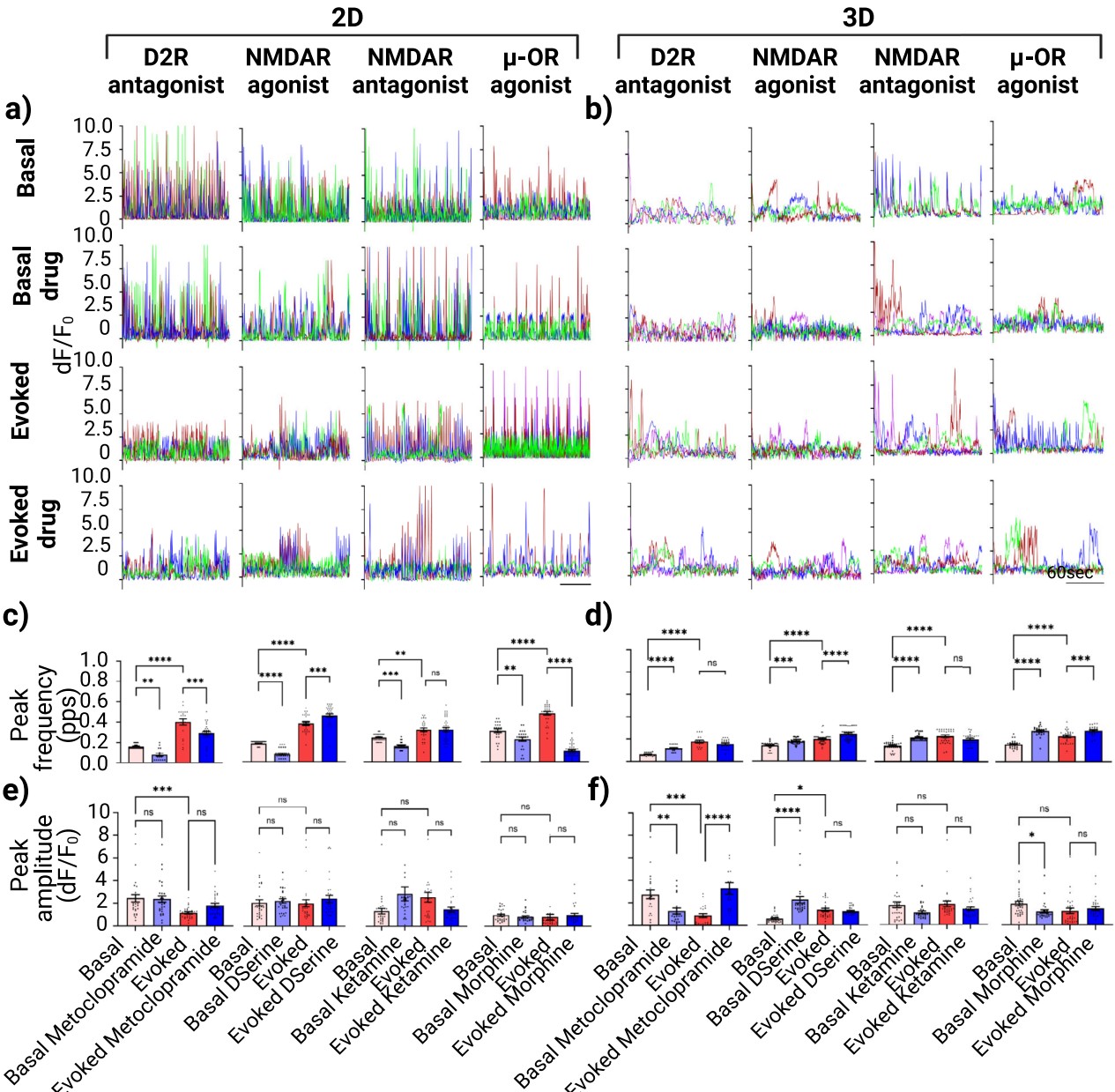

**Fig. 5 Pharmacological modulation of image-based single cell calcium flux for 2D and 3D iDopas/iAstros cellular co-cultures. a**, **b** Example of ROIs from every pharmacological experiment with each physiological variations; Basal, Basal with drug, Evoked, Evoked with drug; for the 2D and 3D co-cultures models, respectively. Quantitative measurements of calcium peak frequency (**c**, **d**) and mean peak amplitudes from each ROIs (**e**, **f**), the spontaneous and evoked calcium flux, for 2D and 3D co-cultures for each drug tested. Minimal of three ROIs from each well and three technical replicates, each with 3 wells per group were plotted as bar graph. Error bar s.e.m., Statistics: Two-way ANOVA with post hoc turkey test between groups.

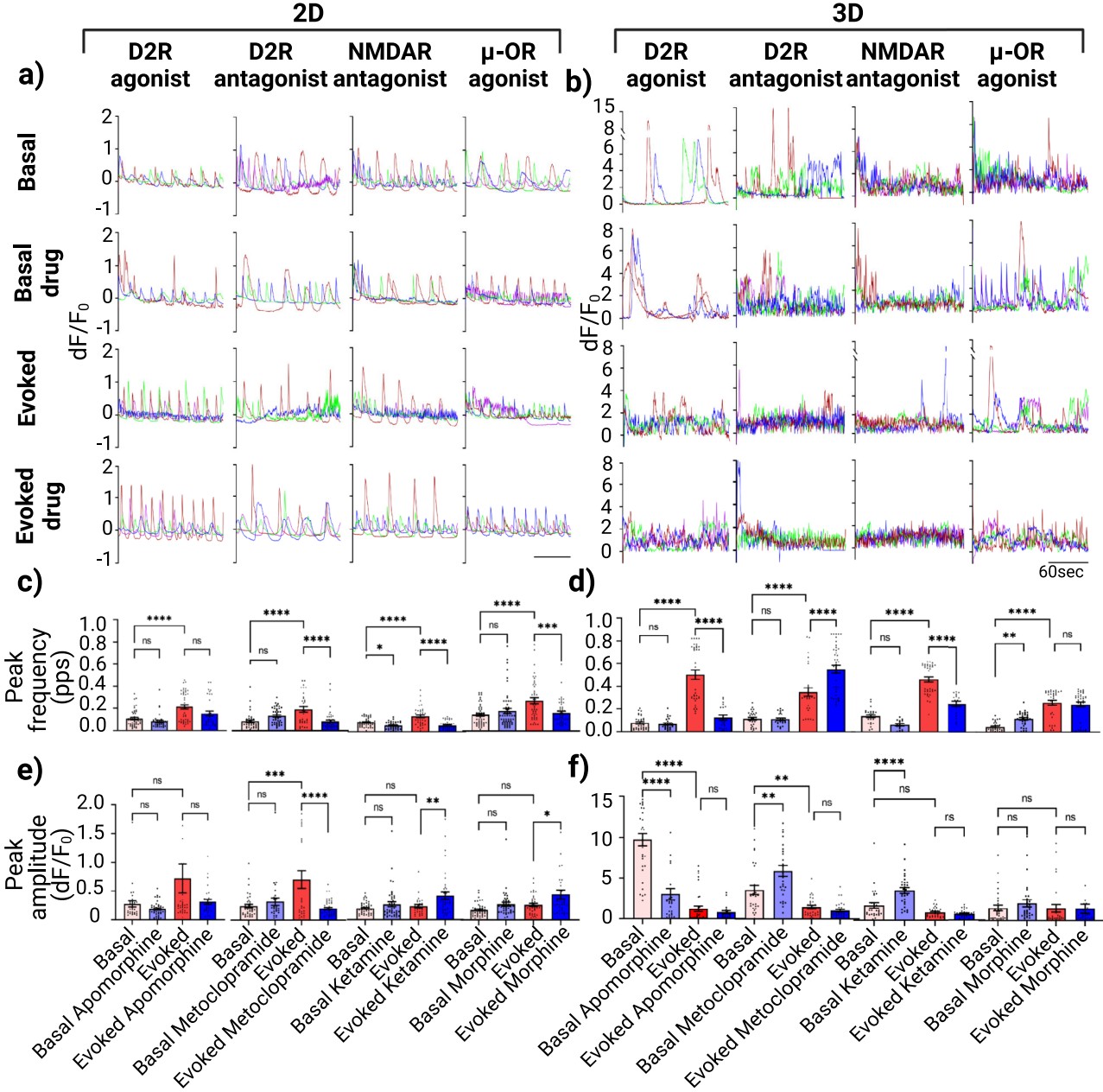

**Fig. 6 Pharmacological modulation of image-based single cell calcium flux for 2D and 3D iGlutas/iAstros cellular co-cultures. a, b** Example of ROIs from different pharmacological treatments in each experimental condition: Basal, Basal with drug, Evoked, Evoked with drug; for the 2D and 3D co-cultures models, respectively. Quantitative measurements of calcium peak frequency (**c, d**) and mean peak amplitudes from each ROIs (**e, f**), the spontaneous and evoked calcium flux, for 2D and 3D co-cultures for each drug tested. Minimal of three ROIs from each well and three technical replicates, each with 3 wells per group were plotted as bar graph. Error bar s.e.m., Statistics: Two-way ANOVA with post hoc turkey test between groups.

models as clinically predictive assays for drug testing. One of the main goals of our work was to produce 3D neural organotypic models that were functional at a neuronal density similar those found in humans and benchmark their physiological activities and pharmacological responses to in vivo data and their 2D counterpart. Postmortem study on human brain suggests that the average neuronal density of visual cortex is 40 K/mm$^3$ [40], likewise the cell density of prefrontal cortex and cerebellar cortex are 34 K/mm$^3$ to 50 K/mm$^3$ [41,42]. Furthermore, the human brain consists of 15–20% of extracellular space of which 20–30% is extracellular matrix, mostly composed of proteoglycans, hyaluronan and small linker proteins submerged in interstitial fluid that closely resembled of cerebrospinal fluid[43–45]. Natural biopolymers like

fibrinogen[46,47], collagen[48], laminin[49] have been used to recreate extracellular neural matrices in vitro. These biopolymers provide the neurons with a support and porosity to adhere, proliferate, form extensions, and if necessary to migrate[50], and allow for the diffusion of nutrients for growth and neuro-chemical for network communication. In this study, we mixed fibrinogen and thrombin to generate interconnected protofibrils as a scaffold[51] with gelatin as a molecular derivatives of collagen for in vivo relevance to structural integrity and cellular growth support[52] and laminin to help improve axonal growth[53] and neuronal viability. This hydrogel composition provided an extracellular scaffold for the neurons to grow and form functional neural networks at 100-fold lower neuronal densities as compared to the average neuronal

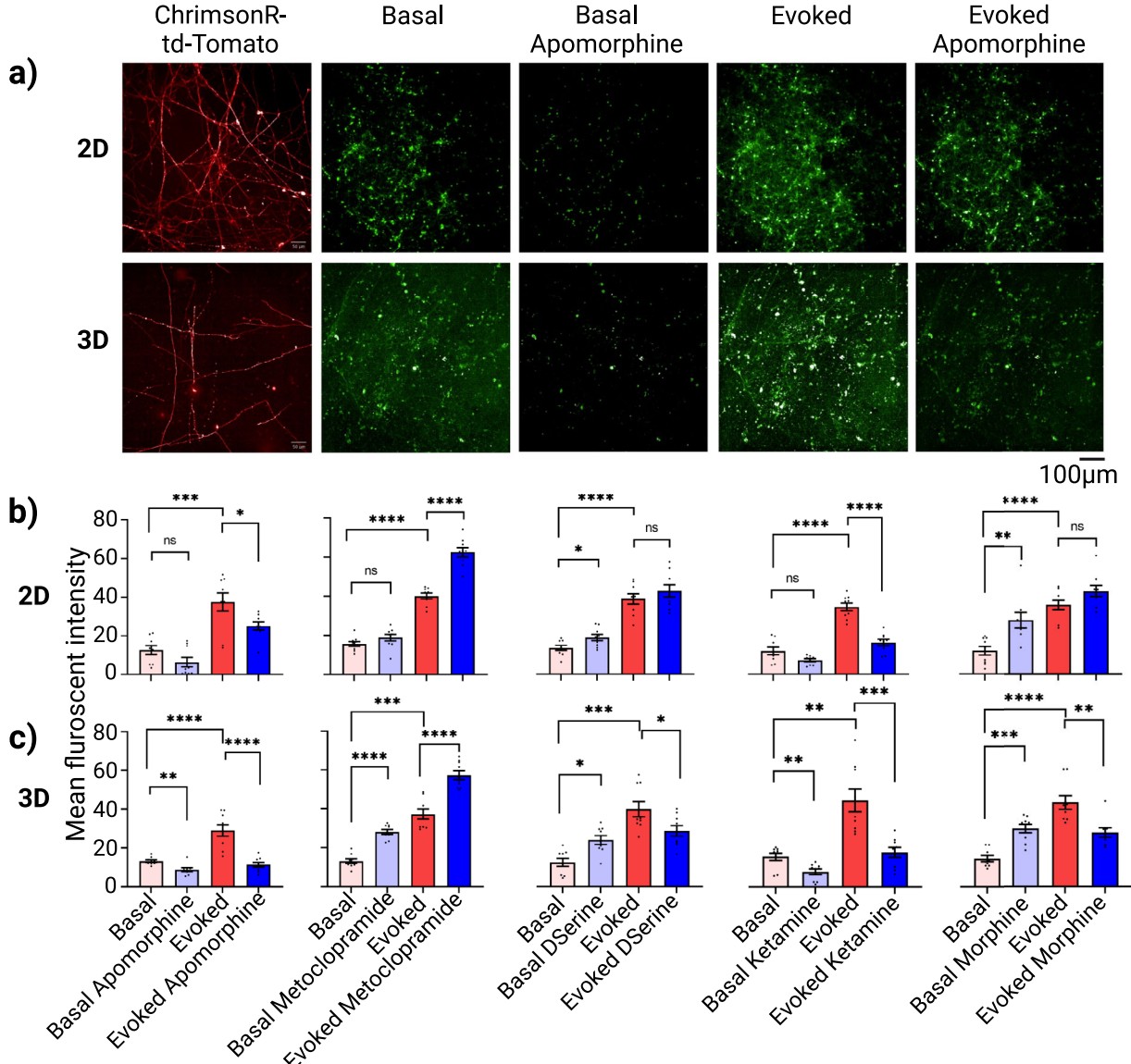

**Fig. 7 Image-based quantitation and pharmacological modulation of dopamine release by genetically encoded dopamine sensor in iDopas/iAstros 2D and 3D co-cultural model.** The genetically encoded dopamine biosensor, hSyn-dLight1.2, was used to quantitate release of dopamine neurotransmitter at the synaptic clefts in the 2D and 3D gel neural co-culture models. **a** Representative images of expression of ChrimsonR via td-Tomato fluorescence, dopamine neurotransmitter release via dLight1.2 as green puncta at the synapses, for both basal and optical stimulation, and with or without Apomorphine, the D2-receptor agonist treatment. **b**, **c** Quantitative estimation of mean fluorescent intensity from multiple fields of views for 2D system ($n = 9$ wells) and from entire z-stack along with multiple fields of views for 3D system ($n = 9$ wells) were accumulated and presented as bar plots. Quantitation of dopamine release by measuring green fluorescence from dLight1.2 biosensor in 2D (**b**) and 3D (**c**) models, after 30 min min treatment with apomorphine, metoclopramide, D-serine and ketamine at 10 μM, and morphine at 15 μM. Scale bar 50 μm. Error bar s.e.m. three technical replicates, with $n = 3$ wells each. Statistics: Two-way ANOVA with post hoc turkey test between groups.

density of 2000K/mm$^3$ in neuronal spheroids[54]. We were able to mix 25 K/μl of human iPSC derived either dopaminergic or glutamatergic neurons, combined with human iPSC derived astrocyte in 1:5 ratio. The confocal microscopic images revealed that the height of our 3D gel-based structures was 100–300 μm in 384 multiwell plate format, thus estimating a neuronal density of around 95K-100K/mm$^3$ in our 3D fibrin gel-based tissue mimic, which is only 2–3-fold higher than the average neuronal density of human brain, still demonstrating physiologically relevant 'in-network' functional modulation. 3D rendered images of the antibody staining of our 3D gel-based model provided the evidence of long axonal projections, spreading both horizontally and vertically, with the expression of neuronal selective markers like

TH on dopaminergic neurons and Glut1 on glutamatergic neurons and adequate expression of postsynaptic marker PSD-95 showing strong synapse formation.

The second goal of developing the 3D gel-based in vitro model was to achieve a human native-like physiological system that can be used as an assay platform for pharmacological testing in a reproduceable manner using functionality driven assays. For that, the used gel-based matrix provided enough optical clarity to enable fluorescence confocal microscopy without the need for optical clearing procedures. Our image-based calcium flux assay using genetically encoded GCaMP6f biosensors allowed us to probe in real time into the 3D neuronal network at the single cellular level and study the calcium activity within the network

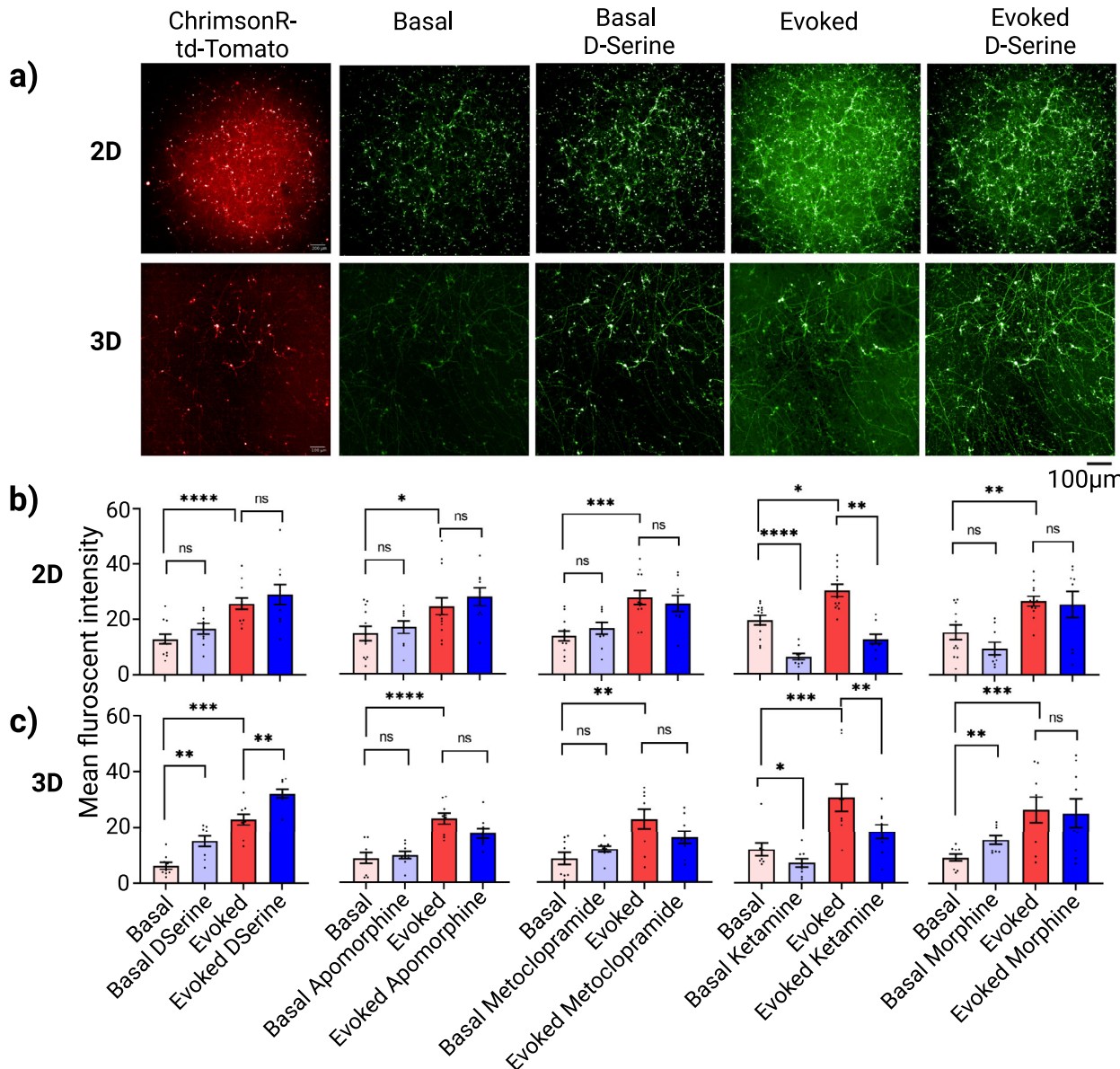

**Fig. 8 Image-based quantitation and pharmacological modulation of glutamate release with a genetically encoded glutamate sensor in iGlutas/iAstros 2D and 3D co-cultural models.** The genetically encoded glutamate biosensor, iGluSnFr was used to quantitate release of glutamate neurotransmitter in both 2D and 3D gel neural co-culture models. **a** Representative images of expression of ChrimsonR (red) via td-Tomato fluorescence, glutamate neurotransmitter release via iGluSnFr (green) with affinity towards Glut1 transporter of glutamatergic neurons, which locates multiple release sites along the neurite's length, for both basal and optical stimulation, and with or without D-serine, the NMDA-receptor agonist treatment. **b** and **c** Quantitation of mean fluorescent intensity from multiple fields of views for 2D system ($n = 9$ wells) and from entire z-stack along with multiple fields of views for 3D system ($n = 9$ wells) were accumulated from each experimental group and presented as bar plots. Quantitation of glutamate release by measuring green fluorescence from iGluSnFr biosensor in 2D (**b**) and 3D (**c**) models, after 30 min treatment with D-serine, apomorphine, metoclopramide and ketamine at 10 µM, and morphine at 15 µM. Scale bar 50 µm. Error bar s.e.m. three technical replicates, with $n = 3$ wells each. Statistic: Two-way ANOVA with post hoc turkey test between groups.

and exhibited its strong attenuation variations after treatment with various receptor specific pharmacological agents. To stimulate neuronal networks in a controlled manner, we included red-shifted opsins, ChrimsonR via AAV transfection. We also used genetically encoded biosensors to assess the 'in-network' release of neurotransmitter. The changes in fluorescent intensities of bio-engineered Dopamine-receptor biosensor, dLight1.2[31] in dopaminergic neuronal network model, and similarly from bio-engineered Glut1-transporter, GluSnFr biosensor[32] in Glutamatergic neuronal network provide us the network modulation information at the synaptic label with physiological and pharmacological perturbation. Our data demonstrated that optogenetics activation of the engineered 3D neural systems produced increases in signal similar and in many cases stronger than the corresponding 2D cellular models. This data validated our approach of using transduced biosensors as robust assay readouts in 3D organotypic models. Furthermore, the flexibility of our biofabrication and image-based assay protocol should allow us to incorporate many more biosensors as readouts in future depends on the neuronal network specificity and synaptic connectivity.

Once we established a protocol for the reproducible biofabrication of 3D neural coculture systems with spontaneously formed neural networks and autonomous synaptic connections, we assessed their physiological and pharmacological relevance by determining the functional responses of the signals for the different biosensors to drugs targeting relevant neural receptors. In our current study we observed relevant differences in pharmacological responses to different treatments between 2D vs 3D neuronal co-culture models. Evidently, 3D gel-matrix altered the chemotherapeutic agents' uptake and retention of growth factors by creating a diffusion gradient and homeostasis surrounding the cells, mimicking its native in vivo environment better, thus changing the cell signaling, pharmacology or toxicology responses[55,56]. Although differences between 2D vs 3D cellular systems have been shown, little work has been published to show the biological and cellular reasons for those differences, which could include changes in receptor expression levels, activation state of signaling pathways, gene expression, diffusion and caging effects, and others. In the work presented here, we observed differences in physiological and pharmacological responses between 2D and 3D neural co-culture models, and work is in progress, using a combination of approaches including transcriptomics and proteomics, to investigate and explain the differences observed.

Deep brain stimulation data from Parkinson patients[57], and electrode recording data from substantia nigra and ventral tegmental area' dopamine neurons of rat midbrain[58] showed that treatment with apomorphine, a D2 receptor agonist, slowed down the dopamine neuronal spontaneous firing, with increasing bursting firing pattern. In our study, we observed reduction in basal calcium peak frequency after treating the 3D gel-based model with apomorphine. We also observed significant reduction in mean calcium peak amplitude, which has been shown to accompany increase bursting pattern[59] of dopaminergic neurons observed in vivo. These pharmacological effects of apomorphine in the 3D iDopas/iAstros co-culture were not observed in the corresponding 2D monolayer iDopas/iAstros coculture. We also observed a reduction in the released dopamine neurotransmitter at synapse from its basal level after treating with apomorphine in the 3D model, which is also in agreement with the findings in human studies by positional emission tomography[60] where they showed that apomorphine inhibited the presynaptic dopamine levels. Many animal behavioral studies have established that optical manipulation creates perception without sensory input by showing whisker movement[61], creating fear memory[62], increasing social interaction[63] and other physiological effects. We measured a significant increase in calcium peak frequency and in the levels of released dopamine neurotransmitter at the synaptic cleft upon optogenetic stimulation. These optogenetically evoked effects were again robustly suppressed by apomorphine treatment in our 3D fibrin gel iDopas/iAstros model, while in the 2D culture, the effect of apomorphine was opposite in peak frequency and not as robust in reducing levels of dopamine.

We treated our 2D and 3D dopaminergic neuronal models with the D2-receptor antagonist, metoclopramide, commonly used for nausea, vomiting and migraine, but can also induce encephalopathy and reduce dopamine efficiency in Parkinson patients[64,65]. While the 2D dopaminergic monolayer culture failed to show a relevant in vivo pharmacological phenotype, the 3D fibrin gel-based neuronal network demonstrated increased in calcium peak frequency and dopamine release, which are effects opposite to those seen for apomorphine, a D2-receptor agonist. Interestingly, metoclopramide suppressed the evoked calcium peak frequency, while still increasing levels of evoked dopamine neurotransmitter release.

D2 receptors are also expressed and modulate the activity of excitatory glutamatergic neurons[66,67]. We therefore tested the functional activity of the D2-receptor agonist and antagonist on the activity of the 3D gel-based iGlutas/iAstros model. Published data suggests that without the afferent connection with dopaminergic neurons, like in addiction circuitry in human ventral tegmental area[68], D2-receptor agonist or antagonist do not have any significant effect on excitatory neurons[69,70]. In agreement, we did not observe significant modulation in terms of basal calcium peak frequency or Glutamate neurotransmitter released after the treatment with apomorphine or metoclopramide. Argumentatively, catalepsy study on rat suggests that metoclopramide can cause noticeable sensitization to NMDA-receptor medicated catalepsy in striatum and frontal cortex via increase in glutamatergic transmission in dose dependent manner. This partial agonistic effect of D2-receptor antagonist in cataleptic rat model was explained by conditional increased in NMDA receptor density after pharmacological intervention[71]. Even though we do not have enough evidence to suggest changes in NMDA receptor density in our models, the increase in calcium peak amplitude at basal and in evoked peak frequency modulation after metoclopramide treatment in our 3D iGlutas/iAstros model suggests future mechanistic studies. Similarly, for the suppression of basal calcium peak amplitude and evoked calcium peak frequency after apomorphine treatment observed in 3D iGlutas/iAstros model, it was observed in a study on the motor cortex function of healthy humans, that apomorphine inhibited transcranial magnetic stimulation evoked motor potentiation in a dose dependent manner[72].

The pharmacological effects of an NMDA-receptor agonist and an antagonist, D-serine and ketamine, respectively, on our 3D neuronal co-cultural model were also in agreement with data published using in vivo models[73–76]. As expected, in the glutamatergic 3D co-culture model, D-serine and ketamine treatments had opposite effects: while D-serine consistently increased both basal calcium peak frequency and glutamate levels, ketamine robustly reduced the evoked calcium peak frequency and glutamate levels. In the 3D dopaminergic system, D-serine increased both the basal or evoked calcium peak frequency, while ketamine increased basal peak frequency and had no effect on the evoked signal. In addition, D-serine significantly increased levels of basal dopamine while decrease labels of evoked dopamine in this co-culture system. Ketamine significantly reduced levels of evoked dopamine release. Collective review and meta-analysis of dopamine measurements in rodents, non-human primates and human brains following acute as well as chronic application of ketamine shows increase dopamine levels in cortex, striatum, and nucleus accumbent. But the chronic application of ketamine showed constant rise in dopaminergic neuronal activity, but the actual changes in dopamine levels were not consistent[77–80]. This inconsistencies in long term detection of dopamine levels upon ketamine treatments could be explained by the presence of dopamine active transporter (DAT) on dopaminergic neurons which might pump the dopamine out of the synaptic cleft into the cytosol[81]. Even though our current 3D fibrin gel iDopas/iAstros model did not have neuronal circuitry level complexity to explain mechanistic differences to in vivo results, it showed an increased basal calcium activity with ketamine even with a reduction in overall dopamine levels. The reduction in dopamine levels by ketamine was even more pronounced after optical stimulation, assumedly endogenous close-loop signaling pathway modulate dopamine clearance in response to physiological demand as shown previously[82]. We backed up our explanation with our D-serine treatment group data, where a modest increase in basal GCaMP did not trigger the dopamine reuptake mechanism[83], displaying increase dopamine level. Following the evoked with

D-serine dopaminergic surge, we noticed significant reduction in dopamine measurement at the synaptic cleft. In the future, it would be interesting to increase the complexity of the design of the neural circuit with two neuronal subtypes or even assessing dopamine reuptake blockers. Increasing evidence suggests dysregulation of this mechanism in dopamine related psychiatric disorders or even in opioid addiction, and it could potentially be a biomarker for disease etiology[84].

μ-opioid receptors are expressed in both of our human iPSC derived glutamatergic and dopaminergic neurons and we were interested in investigating the effects of μ-OR agonists related to addiction. As previously described in the literature, we observed that morphine positively altered the spontaneous dopaminergic activity[85]. Even though the effect of morphine and ketamine on the dopaminergic co-culture within the 3D matrix were similar, they might potentially regulate dopamine levels by different mechanisms. Contrary to ketamine treatment, morphine treatment increased basal dopamine levels, while reducing the evoked levels in our 3D model. In the DAT-knockout mice, which exhibit increased rewarding properties by expressing less dopamine at the ventral tegmental area, morphine was able to stimulate the dopaminergic network to increase extracellular dopamine levels[86]. The presence of both NMDA and μ-opioid receptors on the iDopas used, highlights the intertwingle effects of drugs in the same system, which also suggest that in the future it will be relevant to mix multiple neuronal subtypes to form more complex circuitry by using our current fibrin gel 3D matrix and our newly developed image based spatiotemporal neuronal dynamics analysis method. Rodent studies have also revealed that μ-opioid receptor agonists including morphine increase glutamate release by inhibiting GABAergic interneurons at the ventral tegmental area[87]. In alignment with these results, we also saw significant increases in glutamate levels in the iGlutas/iAstros coculture, although we did not combine any inhibitory neurons in the present system, but it has been predicted that the enhanced derivation of hiPSC glutamatergic neurons also generate GABAergic neurons in the mix[88].

Overall, our data demonstrates that the creation of functional neural networks in a 3D neural co-culture system that respond robustly to optogenetics stimulation using fluorescence biosensors of calcium activity and neurotransmitters levels. Pharmacological responses were in most cases in agreement to those reported in the literature in vivo and in many cases different from those seen using the corresponding 2D neural models. Work is on progress to understand these differences at the molecular level. The 3D gel-based neural model described here appears therefore to be a step forward towards creating physiological relevant functional neural models, and it is amenable to image-based high content screening for therapeutics development. The protocols described here are the starting point to enable the use of bioprinting methods to create more complex neuronal circuits with spatially separated cell specific population to further assess pharmacological effects in a more native-like neural system.

## Methods

### Preparing the plate surface for cell seeding
The 384 well plate (CellCarrier-384 Ultra, PerkinElmer, USA) surface for 2D neuronal co-culture, were prepared by incubating the plate with 30 μl of 0.01% Poly-L-Ornithine hydrochloride (PLO, Sigma, P2533) in PBS, for 1 h at room temperature, followed by the 3 times rinsing with DPBS. Then the plates were coated with 30 μl of 3.3 μg/ml Laminin (Invitrogen, 23017-015) solution in PBS, at 37 °C for 1 h or at 4 °C for overnight. For 3D neuronal co-cultures, the same 384 well plates were treated with oxygen plasma (5cc/min, 10 min) just before the cell seeding. This treatment increased the surface hydrophilicity of the plate and confirmed the adherence of the fibrin gel cocktail with the well surface[89].

### Preparing the mediums and the cells for the neuronal cell culture
Three different Cellular Dynamics International (CDI) (Fujifilm, USA) maintenance mediums, Dopa-medium, Gluta-medium, Astro-medium were prepared for respective cell seeding for Dopaminergic, Glutamatergic neurons and astrocyte. One physiological activity enhances medium, BrainPhys neuronal medium (STEMCELL Technologies, 05790) combining with Bardy's composition[90] were used for maintenance following CDI protocols. Bardy's composition was the combination of 100-fold dilution of Dibutyryl-cyclic-AMP sodium salt (Tocris, 1141) and 1000-fold dilution of L-ascorbic acid (Tocris, 4055).

Post-mitotic human midbrain iPSC derived Dopaminergic (iCell DopaNeurons,01279, Cat#R1032, C1028; Lot#102224, 105288, 102614), Glutamatergic (iCell GlutaNeurons,01279 Cat#R1034, C1033; Lot#105449, 104910, 105905) neurons and Astrocytes (iCell Astrocytes Kit,01434 Cat#R1092, Lot#104345, 105136, 105152, 105337) were procured from CDI (Fujifilm, USA) in $5 \times 10^6$ cells, $6 \times 10^6$ cells and $1 \times 10^6$ cells per vial, respectively, in form of flash frozen and stored in dimethyl sulfoxide (DMSO) in liquid nitrogen. Both neurons' cell types were derived from the same healthy human donor whereas the astrocytes were derived from different healthy donor (according to the data sheet from Fujifilm Cellular Dynamics, Inc, USA). Required amount of cryovials according to the experimental need were thawed out in 37 °C water bath for 1 min without swirling. Then the cryovials were sprayed with 70% ethanol and placed inside the biosafety cabinet for further preparation. The content of each cell vial was then separately and gently transfer to the sterile 50 ml centrifuge tube using 1 ml pipettor and suspended with 10 ml of respective medium. The cell suspensions were then centrifuged at 380 g for 5 min and the supernatant were carefully aspirated without disturbing the cell pellets at the bottom of each tube. Each cell pellets were then resuspended into 4 ml of respective medium for cell viability counting using equal volume of trypan blue exclusion method in Thermo Countess II an automated cell counter from Invitrogen. Over 55% viability for the neurons and over 95% viability for the astrocytes were set as cutoffs for further use in the experiments. Both for 2D and 3D model of neural co-cultures, each type of neurons was seeded with astrocytes in a 5:1 ratio by homogeneous mixing.

### Co-culture of iPSC derived human Dopaminergic (iDopas) or Glutamatergic (iGlutas) neurons with astrocytes (iAstros) in 2D and 3D models
For the 2D co-cultures, the coating laminin solution was aspired from the precoated 384-well plate just before the cell seeding. $33 \times 10^3$ cells of iDopas or iGlutas neurons per well were seeded independently along with homogeneously mixed $6.6 \times 10^3$ iAstros per well for our 2D neuronal co-culture model. The wells were then filled with 50 ul of respective (CDI iDopas or iGlutas) medium and cultured for 2 days in cell culture incubator at 37 °C with 5% $CO_2$. At day 2, the respective neuronal medium was 100% replaced with combination of BrainPhys and Bardy supplements as the maintenance medium for better network development and functional assay detection.

For 3D neuronal culture, the cells were resuspended at $25 \times 10^3/\mu l$ density together with the astrocytes (5:1) in a fibrinogen gel cocktail. The fibrinogen gel cocktail was prepared by combining 2.5 mg/ml Fibrinogen (Sigma F3879) along with 60 mg/ml of Gelatin (Sigma G1890) and 1 μg/ml of Laminin (Invitrogen 23017-015) in PBS. For the 3D neural co-culture models, a gel made up of fibrinogen-thrombin cross-linked fibrin polymer, laminin, and gelatin supported the viable neuronal and astrocytes growth (Supplementary Fig. 1) and enabled the formation of neuronal extensions. We also tested the cell viability of our 3D model using an ATP-based 3D cell viability assay (Supplementary Fig. 1) and uncovered that the addition of 1 μg/ml of laminin helped promoting the axonal growth as literature suggested[35,36] into our 2.5 mg/ml of fibrinogen gel, also improved the viability of our neuronal coculture system as a whole by almost 2-fold (t-test $p < 0.01$ in Supplementary Fig. 1b). 15 μl of fibrinogen gel cocktail with cells were dispensed per well, for 384 well plate. Just before platting, a thrombin solution (Sigma T6884) was added (final 0.5unit/ml) to the cell+fibrinogen cocktail bioink and left for 30 min inside the biosafety hood at room temperature, to initiate fibrinogen crosslinking to form a gelatinous/semisolid 3D matrix. Each well then was filled up with 60 μl of respective neuronal medium and incubated at 37 °C with 5% $CO_2$ for 2 days in cell culture incubator. The flow diagram of developing 2D and 2D neuronal coculture model has given in Fig. 1b.

At day 2, for both 2D and 3D cultures, the respective neuronal medium was 100% replaced by the maintenance medium (BrainPhys + Bardy supplement). The AAV-mediated biosensor and opsin, each at a MOI of $10^6$ viral genome/cell, were delivered to the 2D and 3D co-culture in this maintenance medium. Initially, we performed titration experiments with AAV9/AAV5 serotypes to determine the optimal multiplicity of infection (MOI) level in the iPSC derived human neuronal culture (iDopas + iAstros) in 2D, 96 well microtiter plates. We transfected the neuronal 2D co-culture with increasing concentration ($1 \times 10^4$ vg/cell, $1 \times 10^5$ vg/cell, $2 \times 10^5$ vg/cell and $1 \times 10^6$ vg/cell) of pAAV9-Syn-ChrimsonR-tdTomato (Supplementary Fig. 2a, td-Tomate as in red). After 2 weeks of incubation, we monitored the level of virus-transfection by counting the number of cells expressing td-Tomato, the fluorophore tagged with each viral genome of AAV9-ChrimsonR construct. We counted the number of td-Tomato expressing cells (red cells in merged image of Supplementary Fig. 2a) against the number of non-transfected cells based on the total number of cells forming extensions, as observed on brightfield, and the number of dead cells (appeared as black dots without extensions on the brightfield image). After analyzing five fields of view from each well and combining 3 wells quantification, (Supplementary Fig. 2b), data showed that the $1 \times 10^6$ vg/cell group had the highest ratio of transfected to non-transfected

cells, with modest number of dead cells. Thus, all further experiments were done with $1 \times 10^6$ vg/cell MOI for both AAV9 and AAV5 viral serotypes, for both the 2D and 3D neural co-cultures systems.

Opsin-ChrimsonR (pAAV9-Syn-ChrimsonR-tdT; Addgene 59171) for optogenetic stimulation along with one of the modified G-protein coupled biosensor namely GCaMP6f (pAAV9-CAG-GCaMP6f.WPRE.SV40; Addgene 100836), dLight1.2 (pAAV5-hSyn-dLight1.2; Addgene 111068), iGluSnFr (pAAV5-hSyn-iGluSnFr.WPRE.SV40; Addgene 98929) for functionality assessment were used to transfect the cells. The Adeno-associated virus encapsulated plasmids of red wavelength ($\lambda = 590$ nm) activated opsin ChrimsonR to evoke activity in neurons, and green wavelength ($\lambda = 470$ nm) activated GCaMP6f for detection of spontaneous cellular calcium dynamics, dLight1.2 and iGluSnFr bioengineered G-protein coupled neurotransmitter sensors for detection of dopamine and glutamate (respectively) neurotransmitter release[31,32]. We determined our working MOI of $10^6$ vg/cells for both the AAV serotypes (AAV9 and AAV5) for our human iPSC derived neuronal culture by titration methods, weighing the proportion of transfected/non-transfected/dead cells. The results were shown in supplementary Fig. 2.

The 2D and 3D neuronal co-cultures were maintained for additional 2 weeks with biweekly change of maintenance medium (50%) allow the neuronal culture to fully mature and autonomously form network connection before it underwent testing for viability and functional activity.

**Cell viability assays for neural co-cultures**. Both 2D and 3D neuronal co-cultures model were tested for viability first, after 2 weeks of maintenance and maturation. We performed CalceinAM and Propidium iodine assay for both 2D and 3D cultures, in addition to CellTiterGlo3D assay for the 3D model.

For standard Calcein AM and Propidium iodine assay, 0.5 μM of Calcein AM (C3100MP, Invitrogen) and 1 μM of Propidium iodine (P1304MP) were prepared in 1X DPBS. 30 μl of media were removed from the well, and 30 μl of each dye solution were added at each well. We performed Live/Dead assays in sets of 3 replicate wells per plate. The 2D co-culture plates were then incubated for 30 min, and the 3D neuronal co-cultures were incubated for 1 h for better penetration, at 37 °C in cell culture incubator. After the incubation period, both the plates were imaged under 10X or 40X water objective respectively using the Phenix Opera High Content Screening System (HCSS) (PerkinElmer). FITC ($\lambda_{Ex} = 485$ nm/ $\lambda_{Em} = 535$ nm) and Cy3 ($\lambda_{Ex} = 530$ nm/$\lambda_{Em} = 620$ nm) optical filter were used to image Calcein AM as green and Propidium iodine as red, respectively. For both 2D, 5 fields of view in single plane from each well and for and 3D neuronal culture single field of view from 5 planes from each well were imaged and average mean fluoresce of 3 wells from 3 technical replicates were plotted.

For the CellTiterGlo ATP-based detection assay for of cell viability on the 3D neuronal co-culture, 3 well per group were tested for technical replicates. The maintenance medium from each well was first fully replaced with 60 μl of PBS (1X). Both the 3D neuronal co-culture and the CellTiter-Glo 3D reagent (Promega, G7570) were equilibrated at room temperature approximately for 30 min. 50% of the PBS (30 μL) from each well were then replaced with the CellTiter-Glo 3D reagent and all the content was vigorously mixed by pipetting and stirring on the shaker for 5 min. The plate was then allowed to incubate at room temperature for additional 25 min to stabilize the luminescent signal. Luminescence signal was quantitated as relative luminescence units (RLU) using a ViewLux ultraHTS Microplate Imager (PerkinElmer).

**Immunostaining of neural co-cultures**. For all the neural co-cultures, iDopas +iAstros and iGlutas+iAstros were validated using immunohistochemistry methods. 2D neural co-cultures were fixed with replacing the media with 60 μl of 4% paraformaldehyde in PBS at 4 °C overnight, followed by 3 times washing with 1X PBS. 60 μl of Goat serum (10%) based blocking solution (Invitrogen, 50197Z) was added prior to the primary antibody treatment. 2D co-culture was stained with various combination of primary antibodies as markers of cell identity and synapse, including anti-chicken Tyrosine Hydroxylase (TH) (abchem, ab76442, 1:200 final dilution) with anti-rabbit Glial Fibrillary Acidic Protein (GFAP) (Dako, Z0334, 1:1000 final dilution) and anti-mouse Vesicular Glutamate Transporters 1 (VGlut1) (abchem, ab242204, 1:200 final dilution) with anti-rabbit GFAP (Dako, Z0334, 1:1000 final dilution), anti-mouse Microtubule Associated Protein 2 (MAP2) (Sigma, M4402, 1:250 final dilution), anti-mouse Postsynaptic Density Protein 95 (PSD-95) (Thermofisher, MA1-045, 1:100 final dilution), anti-mouse Neuronal Nuclear Protein (NeuN) (EMD Millipore, MAB377, 1:200 final dilution), and makers of relevant receptors, such as mu-opioid receptor antibody anti-rabbit MOR (abchem, ab10275, 1:200 final dilution). All primary antibodies were added as 60 μl solutions per well by 100% replacement of the blocking solution and incubated overnight at 4 °C, then washed with 1x PBS, followed by incubation with host species selective secondary antibodies (1:500 final dilution, ThermoFishers Scientific), for 1 h at room temperature. After washing with 1x PBS twice, a Hoechst 33342 solution (1:2000 final dilution ThermoFishers Scientific, 62249) was added for 5 min, and the cells were imaged with the Phenix Opera HCSS (Perkin Elmer) and images were taken sequentially in the different channels, using 10X water objectives.

For the 3D neural co-culture model, we performed all the primary antibody staining as described above with some modifications in between steps to improve the penetration through our gel matrix. We used TritonX-100 (Sigma-Aldrich), final concentration 0.2% along with 1X PBS as a solvent throughout our staining process and used plate shaker at 20 RPM at each step of the staining protocol for better tissue permeability. We also increased the incubation time for both primary and secondary staining to overnight at room temperature, on a shaker (20 RPM). Images were taken with a Phenix Opera HCSS (Perkin Elmer) sequentially under the different filter channels, using 10X water objectives.

**Drug treatment**. To study the pharmacological perturbation in our neuronal co-culture both in 2D and 3D model, we selected apomorphine (D2-receptor agonist), metoclopramide (Muscarinic receptor M3 antagonist), D-serine (NMDA receptor co-agonist), ketamine (NMDA receptor antagonist), and morphine (μ-opioid receptor agonist). Each compound except for morphine was tested at 10 μM. Final concentration of morphine was tested at 15 μM. Each compound was tested in three separates biological replicates, each with at least of 3 technical replicates. 10 mM solutions of the compounds in DMSO were diluted as 1:100 in 1x PBS. 5 μl/ well for 2D and 6 μl/well for 3D were added except for morphine. Morphine was added 7.5 μl/well and 9 μl/well respectively by removing equal volume of medium from the well. Drug treatment was done for 5 min and 30 min for 2D and 3D neuronal co-cultures, respectively, before imaging. We assessed that the drugs at the doses tested were not cytotoxic for the cells in both the 2D and 3D neural models by replacing the drug containing medium with fresh medium with no compound and repeating the functional measurements after 24 h incubation.

**Calcium dynamic measurements on 2D neuronal co-cultures**. For the study of functional network activity, we measured calcium flux from the fully matured (2 weeks old) 2D neuronal and astrocyte cocultures, on 384 well plates, using a Calbryte 520AM dye (Fisher Scientific, NC1812243) with the FLIPR Penta High Throughput Cellular Screening System (Molecular Devices). First, 500 μL working solution of Calbryte 520AM was made with HHBS (Hank'sBuffer with HEPES) from 5 mM stock solution. Then 1 μL of 20% Pluronic F127 (ThermoFishers Scientific, P3000MP) was added to the working solution to disperse the acetoxymethyl (AM) esters of the ion indicator dye. 50% of the maintenance medium was replaced from each well with Calbryte 520AM dye's working solution and incubated at 37 °C for 1 h followed by 15 min at room temperature. After the incubation period 100% of the medium with dye solution was replaced with HHBS and placed on FLIPR reading stage for calcium flux measurement. The FLIPR system was pre-calibrated with PerkinElmer 384 Ultra plate parameters following Molecular Devices' working manual. The measurement was taken at 0.6 s interval (1.6 frames/sec) with 50% fluorescent intensity and keeping the threshold cut-off at 8.0. All recording were done by setting the reading plate stage temperature at 37 °C. All the data acquisition were done using the ScreenWorks 5.1 software (Molecular Devices) and post analyzed for the peak properties using Peak Pro 2.0 analysis and statistics software (Molecular Devices). After extracting the calcium peaks information over time (10 min), the peak properties including peak frequency, peak amplitude and area under the peak were plotted in GraphPad Prism 9.0.

Calcium flux measurement were also done using neuronal co-culture transduced with pAAV9.CAG.GCaMP6f.WPRE.SV40 on 384 well plate. The GCaMP6f fluctuations were recorded using the FLIPR with the ScreenWorks 5.1 (Molecular Devices) at 0.6 s interval (1.6 frames/sec) with 50% fluorescent intensity and keeping the threshold cut-off at 8.0. All recording were done by setting the reading plate stage temperature at 37 °C. All the post analysis of the peak properties were calculated using Peak Pro 2.0 analysis and statistics software (Molecular Devices) and plotted in GraphPad Prism 9.0.

For the compounds testing, after acquiring the basal calcium signals from the 2D neuronal culture, we added the compound solution as described above in the 'Drug treatment' section and incubated for 5 min, then started acquiring the calcium signal again keeping every other parameter constant.

**Image based detection of spontaneous and evoked single cellular Calcium activity and its pharmacological modulation on 2D neuronal culture with optogenetics stimulation**. 2D neuronal cocultures were double transfected as described above with pAAV9.CAG.GCaMP6f.WPRE.SV40 and pAAV9-Syn-ChrimsonR-tdT viruses on 384-well plate, and incubated for 2 weeks. After the incubation period, the plate was placed under 10X water objective of Phenix Opera HCSS. At first, the expression of pAAV9-Syn-ChrimsonR-tdT virus was confirmed by taking images of multiple fields of views of each well with selective red laser ($\lambda_{Ex} = 561$ nm and $\lambda_{Em} = 599$ nm) specific for td-Tomato with 30 ms exposure time and 50% laser intensity. Once the ChrimsonR-tdT expression was confirmed via td-Tomato expression, the green laser ($\lambda_{Ex} = 488$ nm and $\lambda_{Em} = 522$ nm) was used to capture the fluorescence intensity of GCaMP6f with 60 ms exposure time of 50% laser intensity. Then we captured the image from the same plane and same field of view simultaneously, stimulating with red laser ($\lambda_{Ex} = 590$ nm) with 40 ms exposure time with 50% laser intensity and recording with green laser ($\lambda_{Ex} = 488$ nm and $\lambda_{Em} = 522$ nm) with 60 ms exposure time with 50% laser intensity. Far red laser was used to stimulate the ChrimsonR opsin expressed on neuronal membrane as our optogenetic stimulation. We quantitated the changes in fluorescence intensity between the spontaneous and evoked GCaMP6f activity using ImageJ

(NIH) image analysis software and plotted the graph in GraphPad Prism 9.0 (Fig. 3a).

To measure live calcium flux over time, we used the Opera Phenix HCSS (10X water objective) and acquired consecutive images for 180 s with 0.6 s interval (1.6 frames/second) keeping all the above-mentioned excitation and emission filter unchanged. We kept the same GCaMP6f acquisition rate as our FLIPR calcium flux study to do systematic comparison of the peak properties. We repeated that same image acquisition overtime for evoked GCaMP6f activity too with simultaneous excitation of far-red laser ($\lambda_{Ex} = 590$ nm). The process of single cellular calcium dynamic measurement was mentioned in data processing section. After acquiring the spontaneous and evoked GCaMP6f activity over time, we added the drugs on those culture wells and incubated for 5 min, set of 3 wells with 3 technical replicates for each drug and repeat the exact same acquisition process for both spontaneous and evoked activity consecutively. We also performed the peak detection and analysis process using LC-Pro plugin of ImageJ and OriginPro9.0 software keeping every other parameter constant.

**Image based detection of single cellular Calcium dynamics of neuronal culture network in 3D gel-based model with optogenetics and pharmacological modulation.** We developed and validated an assay of single cell calcium flux detected based on fluorescence imaging for our 3D gel-based neuronal coculture model. At day two of culture in the microtiter plate, the cells were transduced with a cocktail of pAAV9.CAG.GCaMP6f.WPRE.SV40 and pAAV9-Syn-ChrimsonR-tdT viruses as described above and cultured for 2 weeks. After the incubation period the fluorescence signal was measured with the Phenix Opera HCSS, using 10X water objective with confocal mode. First, we performed a quick scan for each well through the multiple fields of view covering multiple planes to find the optimal plane/field of view with maximum number of GCaMP6f transfected cells under single focal plane. Then for baseline calcium activity measurement, we captured the time series images for 180 s with 1.6 frames/second acquisition rate from that selected plane of each well using green laser ($\lambda_{Ex} = 48\ 8$ nm and $\lambda_{Em} = 522$ nm) with 60 ms exposure time and 50% intensity. Then we repeated the same time sequence acquisition with simultaneous stimulation of ChrimsonR opsin with far red laser ($\lambda_{Ex} = 590$ nm), 40 ms exposure time and 50% laser intensity. We also repeated the same time series experiments for basal and evoked calcium flux sequentially, after 30 min incubation of individual drug compounds.

**Data processing for single cellular calcium dynamics.** The acquisition plane was determined by the satisfactory expression of ChrimsonR via td-Tomato fluorescence. After the selection of imaging plane, a series of 300 time-lapsed images were captured over a period of 180 s with 0.6 s interval by Harmony v5.1 software (PerkinElmer). The time-lapse images were stacked to provide maximum intensity projection and transferred to ImageJ image processing software. Then ROIs were automatically selected centering on the neuronal soma by the 'LC-Pro' plug-in[91] in ImageJ by providing the ROI diameter (pixel) 30, frame rate (fps) 1.6 and for intensity cutoff threshold with p-value 0.05. Finally, the raw intensity values of the calcium signal were extracted from selected ROIs. For the analysis, each ROIs data were then transferred to 'Origin-Pro 9.0'(OriginLab). Single neuronal dynamics were quantified via the calcium signal peak frequency and peak amplitude. The peaks were detected over time by 'positive maximum intensity peak finding method' using 'batch peak processing algorithm', with second derivatives of individual ROI's raw fluorescence intensities (r.f.u.) as a change in fluorescence intensity from initial (dF)[92]. Because of sparsity of event's appearance in our 3D system and to improve the single to noise ratio, we used second derivatives of r.f.u. After thresholding the base line, the estimation of initial fluorescence level ($F_0$) for each ROIs was calculated by averaging the r.f.u. over full acquisition length and then the amplitude was normalized to $dF/F_0$ for respective ROIs. We did not used initial image frame's mean fluorescent intensity as our $F_0$, because in both of 2D and 3D model on every image frame we had GCaMP6f activated fluorescence intensity from cells to some extent. We never got completely 'no signal' in any frame. The oscillations of calcium waves ($dF/F_0$) from 3–4 example traces from each experimental group were presented in Fig. 4b for 2D and Fig. 4e for 3D system. The peak frequencies (peak per sec) and mean peak amplitudes from each ROIs, minimum of 3 ROIs per well from 3 biologically replicated wells per plate and 3 technical replicated plates per group were plotted as bar graph. We repeated the same analysis for each drugs treated group.

**Image-based detection of spontaneous and evoked neurotransmitters (Dopamine and Glutamate) release from respective neuronal co-culture of 2D and 3D model.** Neuronal and astrocytes co-cultures were double-transfected with pAAV9-Syn-ChrimsonR-tdT and either pAAV5-hSyn-dLight1.2 (Addgene 111068) or pAAV9.hSyn.iGluSnFr.WPRE.SV40 (Addgene 98929), as described above, and incubated for 2 weeks. Fluorescence signal was measured with an Opera Phenix HCSS using 10X water objective in epifluorescence mode for 2D and 10X water objective in confocal mode for 3D co-culture. As described above, first, the expression of pAAV9-hSyn-ChrimsonR-tdT virus was confirmed by taking images of multiple fields of views of each well with selective red laser ($\lambda_{Ex} = 561$ nm and $\lambda_{Em} = 599$ nm) specific for td-Tomato with 40 ms exposure time and 50% laser intensity. Once the ChrimsonR-tdT expression was confirmed via td-Tomato

expression, the green laser ($\lambda_{Ex} = 488$ nm and $\lambda_{Em} = 522$ nm) was used to capture the fluorescence intensity of the biosensors with 100 ms exposure time of 50% laser intensity. For the optogenetics stimulation, we captured the image from the same plane and same field of view by simultaneously stimulating with far red laser ($\lambda_{Ex} = 590$ nm) with 40 ms exposure time with 50% laser intensity and recording with green laser ($\lambda_{Ex} = 488$ nm and $\lambda_{Em} = 522$ nm) with 100 ms exposure time with 50% laser intensity. We quantified the differences in fluorescence intensity between the spontaneous and evoked released neurotransmitter using ImageJ (NIH) image analysis software and plotted the graph in GraphPad Prism 9.0. We repeated the same spontaneous/basal and evoked neurotransmitter released protocol after treating our 2D and 3D neuronal co-culture (5 min and 30 min respectively) with our pharmacological agents individually.

**Statistical analysis.** Each bar graphs were plotted in GraphPad Prism 9.0 as mean ±s.e.m. across 4 experimental groups, (i) Basal, (ii) Basal after respective drug treatment, (iii) Evoked, (iv) Evoked after corresponding drug treatment. Each experiment was repeated 3 times as biological replicates, and for each biological replicate, 3 wells were included for each treatment condition, as technical replicates. For measurements of single cellular calcium dynamics, each dot in the bar plots represents the '$n$' = number of ROIs or cells per well. 3–5 cells from each well, autonomously picked by the LC-Pro '*ROI measurement algorithm*' with maximum change in intensities over the series of 300 time-lapsed images for a period of 180 s, were selected for measurements. For the experiments to measure neurotransmitter levels, each dot in the bar plots represents the '$n$' = number of wells. Total of 9 wells were accounted-for (3 biological × 3 technical replicates) per experimental group. Mean fluorescence intensities per well were obtained by averaging the fluorescence intensity of 5 'field of view' of epifluorescence images in the 2D cultures and 5 'field of views' of confocal z-stacks (max-projection) images in the 3D models.

To establish the differences between the groups, we performed Two-way ANOVA test comparing each group's mean with every other group's mean using Turkey statistical hypothesis with 0.05 Family-wise alpha threshold in 95% confidence interval. The statistical differences were represented as * as $p < 0.5$; ** as $p < 0.01$; *** as $p < 0.001$; **** as $p < 0.0001$ and n.s. as no statistical difference.

**Reporting summary.** Further information on research design is available in the Nature Portfolio Reporting Summary linked to this article.

## Data availability
The source data behind the graphs in the paper are available in Supplementary Data 1 and 2. Additional images are available from the authors on reasonable request.

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

## Acknowledgements
This research was supported by the Helping End Addiction Long-term (HEAL) NIH program at the National Center for Advancing Translational Sciences. Authors wanted to thank Dr. David Talmage (NIH/NINDS) for critical reading and comments on the manuscript. Authors thank Min Jae Song (NCATS/NIH) for his help acquiring the high resolution confocal "Featured Image".

## Author contributions
S.K., M.F., M.E.B. conceptualized the study. S.K. and M.F. designed experiments and drafted the original manuscript. S.K. performed all the experiments and data acquisition. M.E.B helped in developing the 3D model. C.E.S. and T.V. helped in calcium dynamics data analysis. M.E.B. and C.E.S. critically reviewed the manuscript. M.F. coordinated the entire project.

## Funding

## Competing interests
The authors declare no competing interests.
