## [Peer Review File · Communications Biology]

Reviewers' comments:

Reviewer #1 (Remarks to the Author):

Summary

The overall goal of this manuscript is to describe and validate novel, high-throughput, 3-dimensional optical assays to measure physiological processes in human induced pluripotent-derived neurons. The authors evaluate two different neuronal populations, one with a dopaminergic phenotype (iDopa) and one with a glutamatergic phenotype (iGlut). Neurons are infected with AAV's expressing various optical tools enabling optogenetic excitation, measuring of calcium currents, and measuring of neurotransmitter release. Neurons are cocultured with astrocytes in wells of a 384-well microplate that are either coated with a thin layer (several microns) of extracellular matrix (ECM) protein (2D cultures) or filled with a thick layer (100-200 microns) of a defined ECM matrix enabling a more 3-dimensional culture organization. The physiological response, including calcium influx and neurotransmitter release, of neurons to various pharmacological manipulations is evaluated in 2D and 3D culture and the results are compared against results previously published in the literature.

Overall Impression

My overall impression of the work is positive. The article is well written with the exception of a few scattered typos. I agree with their assessment of the field and the need for development of the type of tools described in the manuscript. I also think they did a good job of with the initial characterization of this system and attempts to interpret the type of data it produces

Specific Comments

1. The manuscript that is seemingly laid out in the introduction does not really match the experiments that were actually executed. The introduction focuses mainly on the shortcomings of alternative 3D cultures systems, specifically naming the unnaturally high cell density in spheroids and assembloid cultures, insufficient control over cell composition, and a disproportionate amount of short-distance connections relative to long distance connections. The authors assert that creation of a model with lower cell density, more ECM, and the proper composition of cells will create a more physiologically relevant system. However, the data presented in the manuscript largely compares their lower-density culture system to a 2D culture system, not to spheroid or assembloid-based culture systems. They do not vary the seeding density in the system to show distortion of physiology at high density, they do not investigate the creation of short vs long-distance connections, and they do not vary cell composition to show a distortion of physiology and in fact they do not show that astrocytes need to be included at all in this system. So although I agree that it is important to consider all of these features in the design of model system and that their system could potentially recapitulate in vivo physiology more faithfully than other current models, they do not actually demonstrate advancement over these other model systems with the data presented in the manuscript.

2. There are several references to measuring "plasticity" and "network plasticity" in the manuscript. The authors need to provide a definition for this term. I typically understand plasticity to mean permanent, experience-dependent changes in network activity, in that the same input produces a different output before and after the plasticity-induced experience. Showing differences in physiology in the presence of drug does not constitute my understanding of plasticity, it is just a drug-dependent change in physiology.

3. I am concerned about the lack of washout experiments. They record baseline physiology, and then physiology in the presence of various drugs, but they do not then washout the drugs and re-evaluate physiology to determine if any identified effects are purely drug-dependent and return to baseline or if they induce more permanent changes in physiology (which could arguably be plasticity). Controls like this are good practice and may reveal something you didn't know that you didn't know about the assay and I see no mention of them throughout the manuscript.

4. The authors claim "we did not observe any significant modulation both in terms of calcium activity or glutamate neurotransmitter released after treatment with apomorphine or metoclopramide" on the

3D iGluta cell type. Figure 6D and 6F seems to clearly show that the D2R agonist (apomorphine) significantly reduces evoked 3D iGluta spike frequency and basal spike amplitude and the D2R antagonist (metoclopramide) significantly increases evoked 3D iGluta spike frequency and basal spike amplitude. Is this not a direct contradiction? Does it need clarification?

5. The discussion focuses largely on the comparison of the results of the 3D version of the assay to other reports previously published in the literature. The authors acknowledge that this is a difficult comparison because in vivo networks often involve interactions of multiple distinct types of neurons that are not included yet in this system, like GABAergic cells, that play a large role in mediating drug dependent effects at the systems level. But what I find lacking is a discussion of why there are such different, and often completely opposite, results observed from the same exact composition of cells when cultured in 2D vs 3D. This again is the comparison that was actually performed and could have a lot of relevance to basic bioengineering practices but received no mention in the discussion section.

6. The authors should clarify how the statistics were performed and what exactly constitutes each "n" in their ANOVAs. I assume they imaged multiple cells per field, multiple fields per well, multiple wells per experiment, and performed multiple experiments. So is each "n" in the experiment a single cell? The average of all cells in a single FOV? Average of all in a single well? or a single experiment?

Reviewer #2 (Remarks to the Author):

The manuscript proposed the 3D organotypic models composed of iPSC differentiated glutamatergic neurons or dopaminergic neurons and astrocytes. The authors used the gel-based extracellular matrix to incorporate cells in the 3D model. They also showed functional connectivity of these neurons through pharmacological perturbation and optogenetics. Co-culture of various cell types would be beneficial for the functional study of neural circuits. Nonetheless, the referee encourages the authors to provide the novelty of this work and comment on the following aspects.

1. Recently, brain organoids composed of various cell types have been introduced (ex: <https://doi.org/10.1038/s41467-021-24775-5>). The authors need to mention the advantage of the proposed model compared to the brain organoid.
2. There are some previous co-culture platforms that contain neurons with astrocytes (ex: <https://doi.org/10.1016/j.neuro.2018.06.007>). It is highly required to explain the novelty of this work compared to previous platforms.
3. There are also various hydrogel-based 3D cell culture platforms (ex: doi: 10.1038/ncomms14346). Please compare the proposed platform with previous ones.
4. The manuscript is too lengthy. It is strongly required to reduce the length of the manuscript. For example, the authors explained results with numbers. It would be better to specify numbers in Figure captions. Also, some sentences in the Result section can be moved to the Method section.
5. The text in the Figures is too small. Please increase the font size.

Reviewer #3 (Remarks to the Author):

High throughput 3D gel-based neural organotypic model for cellular assays with fluorescence biosensors.

This is a very well written and interesting manuscript and a very enjoyable read. The major claims of the paper were the development of 3D organotypic model based in a fibrin hydrogel with human iPSC derived cells. Such cells are genetically modified to fluoresce in real time for measurements. These novel claims are particularly useful for multiple reasons. Firstly, for reducing the need for animal-based experiments. Secondly, the model can act as an assay that allows for hundreds of experimental statistical repeats without the need for animals and a model that is physiologically more like humans. Thirdly, the genetically modified cells that fluoresce will allow for accurate tracking and monitoring of cells in real-time that is useful in the fields of developmental biology and neurodegenerative diseases.

The conclusions made are original and valid based on thorough experiments and statistical analysis performed. Given the level of detail provided in the methods, a researcher would be capable to reproduce the work. Overall, this paper will benefit the field of neuroscience, developmental biology and potentially regenerative medicine.

I recommend that this manuscript is published as is.

We thank all the reviewers for their valuable comments. We addressed all the points raised by the referees and incorporated the changes in our revised manuscript (highlighted in blue/cyan).

Reviewers' comments:

Reviewer #1 (Remarks to the Author):

Summary

The overall goal of this manuscript is to describe and validate novel, high-throughput, 3-dimensional optical assays to measure physiological processes in human induced pluripotent-derived neurons. The authors evaluate two different neuronal populations, one with a dopaminergic phenotype (iDopa) and one with a glutamatergic phenotype (iGlut). Neurons are infected with AAV's expressing various optical tools enabling optogenetic excitation, measuring of calcium currents, and measuring of neurotransmitter release. Neurons are cocultured with astrocytes in wells of a 384-well microplate that are either coated with a thin layer (several microns) of extracellular matrix (ECM) protein (2D cultures) or filled with a thick layer (100-200 microns) of a defined ECM matrix enabling a more 3-dimensional culture organization. The physiological response, including calcium influx and neurotransmitter release, of neurons to various pharmacological manipulations is evaluated in 2D and 3D culture and the results are compared against results previously published in the literature.

Overall Impression

My overall impression of the work is positive. The article is well written with the exception of a few scattered typos. I agree with their assessment of the field and the need for development of the type of tools described in the manuscript. I also think they did a good job of with the initial characterization of this system and attempts to interpret the type of data it produces

Specific Comments

1. The manuscript that is seemingly laid out in the introduction does not really match the experiments that were actually executed. The introduction focuses mainly on the shortcomings of alternative 3D cultures systems, specifically naming the unnaturally high cell density in spheroids and assembloid cultures, insufficient control over cell composition, and a disproportionate amount of short-distance connections relative to long distance connections. The authors assert that creation of a model with lower cell density, more ECM, and the proper composition of cells will create a more physiologically relevant system. However, the data presented in the manuscript largely compares their lower-density culture system to a 2D culture system, not to spheroid or assembloid-based culture systems. They do not vary the

seeding density in the system to show distortion of physiology at high density, they do not investigate the creation of short vs long-distance connections, and they do not vary cell composition to show a distortion of physiology and in fact they do not show that astrocytes need to be included at all in this system. So although I agree that it is important to consider all of these features in the design of model system and that their system could potentially recapitulate in vivo physiology more faithfully than other current models, they do not actually demonstrate advancement over these other model systems with the data presented in the manuscript.

Ans: As suggested by the reviewer, we have re-written the Introduction to address the comparison of the 3D vs 2D models (Please see page 2-3, highlighted in blue).

2. There are several references to measuring “plasticity” and “network plasticity” in the manuscript. The authors need to provide a definition for this term. I typically understand plasticity to mean permanent, experience-dependent changes in network activity, in that the same input produces a different output before and after the plasticity-induced experience. Showing differences in physiology in the presence of drug does not constitute my understanding of plasticity, it is just a drug-dependent change in physiology.

Ans: We used the word plasticity in the context of certain cells being able to change their phenotype in response to environmental cues. We have now edited the use of the word plasticity only for this context and used the word ‘modulation’ instead for changes in drug-dependent functional phenotypes (highlighted in blue).

3. I am concerned about the lack of washout experiments. They record baseline physiology, and then physiology in the presence of various drugs, but they do not then washout the drugs and re-evaluate physiology to determine if any identified effects are purely drug-dependent and return to baseline or if they induce more permanent changes in physiology (which could arguably be plasticity). Controls like this are good practice and may reveal something you didn’t know that you didn’t know about the assay and I see no mention of them throughout the manuscript.

Ans: Excellent suggestion and we plan to do systematic washout experiments in the future to study drug withdrawal effects. Although detailed washout experiment has not been completed yet, but we did calcium measurement after 24 hrs of drug wash-out for randomly picked treatments group to assess whether there were permanent effects on neural functional and ability to re-activate the systems. As an example of this data, we have added a new supplementary figure from iDopas/iAstros, Apomorphine treated group (Supplementary figure 6, please see page 36), showing how calcium activity resumed in both 2D and 3D co-cultures to pre-treatment levels after 24hr of drug washout. In addition, both 2D and 3D co-cultures were able to be functionally activated with optical stimulated at levels similar to pre-treatment conditions (Page 14, highlighted in blue).

4. The authors claim “we did not observe any significant modulation both in terms of calcium activity or glutamate neurotransmitter released after treatment with apomorphine or metoclopramide” on the 3D iGluta cell type. Figure 6D and 6F seems to clearly show that the D2R agonist (apomorphine) significantly reduces evoked 3D iGluta spike frequency and basal spike amplitude and the D2R antagonist (metoclopramide) significantly increases evoked 3D iGluta spike frequency and basal spike amplitude. Is this not a direct contradiction? Does it need clarification?

Ans: We thank the reviewer for this comment. We unintentionally generalized the effects of drugs in the discussion by stating “we did not observe any significant modulation both in terms of calcium activity or glutamate neurotransmitter released after treatment with apomorphine or metoclopramide”. As the reviewer pointed out, we indeed see effects of apomorphine or metoclopramide of the 3D iGluta cell type. We have now clarified this observation in the Results section in page 14 (highlighted in blue) and discussed the effects of these drugs in parallel to *in vivo* study in more detail (Please see Discussion section, page 17-18, highlighted in blue).

5. The discussion focuses largely on the comparison of the results of the 3D version of the assay to other reports previously published in the literature. The authors acknowledge that this is a difficult comparison because *in vivo* networks often involve interactions of multiple distinct types of neurons that are not included yet in this system, like GABAergic cells, that play a large role in mediating drug dependent effects at the systems level. But what I find lacking is a discussion of why there are such different, and often completely opposite, results observed from the same exact composition of cells when cultured in 2D vs 3D. This again is the comparison that was actually performed and could have a lot of relevance to basic bioengineering practices but received no mention in the discussion section.

Ans: We are indeed very interested in understanding the mechanistic reasons that can explain the different pharmacological responses between 2D and 3D cultures. The experiments that would help answer this question, e.g. scRNAseq, receptor expression and density are planned, we will report them when available because we agree that it would be of high interest to the scientific community. We believe that the scope of this manuscript is to demonstrate that such physiological and pharmacological differences between 2D and 3D cellular models indeed exist, and that the responses observed in 3D models are closer to those seen *in vivo*. We can speculate as to possible mechanistic reasons, and we did that in the Discussion section, including adding some additional references (Please see page 17, highlighted in blue).

6. The authors should clarify how the statistics were performed and what exactly constitutes each "n" in their ANOVAs. I assume they imaged multiple cells per field, multiple fields per well, multiple wells per experiment, and performed multiple experiments. So is each "n" in

the experiment a single cell? The average of all cells in a single FOV? Average of all in a single well? or a single experiment?

Ans: We have clarified the statistics that we performed under the Materials and Method, in Statistical analysis section (Please see page 10, highlighted in blue).

Reviewer #2 (Remarks to the Author):

The manuscript proposed the 3D organotypic models composed of iPSC differentiated glutamatergic neurons or dopaminergic neurons and astrocytes. The authors used the gel-based extracellular matrix to incorporate cells in the 3D model. They also showed functional connectivity of these neurons through pharmacological perturbation and optogenetics. Co-culture of various cell types would be beneficial for the functional study of neural circuits. Nonetheless, the referee encourages the authors to provide the novelty of this work and comment on the following aspects.

1. Recently, brain organoids composed of various cell types have been introduced (ex: <https://doi.org/10.1038/s41467-021-24775-5>). The authors need to mention the advantage of the proposed model compared to the brain organoid.

Ans: We thank the reviewer for the reference and the comment. Indeed, tissue bioengineering technologies are evolving at a very fast pace to solve the issues with each of the current types of 3D organotypic models. As suggested by the reviewer, we have re-written the Introduction to highlight the differences between organoids and biofabricated 3D organotypic models and have highlighted what we think are the advantages of our proposed model (Please see Introduction, page 3, reference 28, highlighted in blue). We are also pointing now that our approach is very versatile, and some of the human brain tissue-derived brain extracellular matrices described in the manuscript referenced by the reviewer are indeed appealing and could potentially be applied to our experimental design to help improve physiological relevance.

2. There are some previous co-culture platforms that contain neurons with astrocytes (ex: <https://doi.org/10.1016/j.neuro.2018.06.007>). It is highly required to explain the novelty of this work compared to previous platforms.

Ans: We thank the reviewer for the reference and the comment, as well. There are indeed very elegant neural co-culture platforms with neurons and astrocytes, and sophisticated spatial arrangements to create functional neural networks. Those are in general very specialized devices which are not designed for the throughput needed for drug screening. Our approach builds on a

capability that will enable the use of bioengineering approaches, like bioprinting, which are now being deployed in many labs, to create versatile and modular biofabrication protocols to build physiologically relevant neural models, together with phenotypic functional assays, in a multiwell plate format to enable drug screening. We have re-written the introduction to discuss this and added references (Please see page 16, reference 49, highlighted in blue).

3. There are also various hydrogel-based 3D cell culture platforms (ex: doi: 10.1038/ncomms14346). Please compare the proposed platform with previous ones.

Ans: We thank the reviewer for the reference and the comment, as well. We have re-written the Introduction to address this comment and incorporated the reference (Please see page 3, reference 27, highlighted in blue).

4. The manuscript is too lengthy. It is strongly required to reduce the length of the manuscript. For example, the authors explained results with numbers. It would be better to specify numbers in Figure captions. Also, some sentences in the Result section can be moved to the Method section.

Ans: Thanks for the comment. We have edited the introduction and results section to shorten the manuscript.

5. The text in the Figures is too small. Please increase the font size.

Ans: Thanks for the comment. We have increased the font size for the text in all figures.

Reviewer #3 (Remarks to the Author):

High throughput 3D gel-based neural organotypic model for cellular assays with fluorescence biosensors.

This is a very well written and interesting manuscript and a very enjoyable read. The major claims of the paper were the development of 3D organotypic model based in a fibrin hydrogel with human iPSC derived cells. Such cells are genetically modified to fluoresce in real time for measurements.

These novel claims are particularly useful for multiple reasons. Firstly, for reducing the need for animal-based experiments. Secondly, the model can act as an assay that allows for hundreds of experimental statistical repeats without the need for animals and a model that is physiologically more like humans. Thirdly, the genetically modified cells that fluoresce will allow for accurate

tracking and monitoring of cells in real-time that is useful in the fields of developmental biology and neurodegenerative diseases.

The conclusions made are original and valid based on thorough experiments and statistical analysis performed. Given the level of detail provided in the methods, a researcher would be capable to reproduce the work. Overall, this paper will benefit the field of neuroscience, developmental biology and potentially regenerative medicine.

I recommend that this manuscript is published as is.

REVIEWERS' COMMENTS:

Reviewer #1 (Remarks to the Author):

The authors have adequately addressed my critique and I support publication of the manuscript. It is great work.

Reviewer #2 (Remarks to the Author):

Thank you for taking the time and effort to address reviewer concerns. The authors have mostly addressed all my concerns and now have provided a much-improved version of the manuscript.